# Nanoparticle-enhanced radiotherapy synergizes with PD-L1 blockade to limit post-surgical cancer recurrence and metastasis

Xin Guan 1,2,3,6, Liping Sun1,2,3,6, Yuting Shen1,2,3,6, Fengshan Jin1,2,3, Xiaowan Bo1,2,3, Chunyan Zhu1,2,3, Xiaoxia Han1,2,3, Xiaolong Li1,2,3, Yu Chen 4✉, Huixiong Xu1,2,3,5✉ & Wenwen Yue 1,2,3✉

Cancer recurrence after surgical resection (SR) is a considerable challenge, and the biological effect of SR on the tumor microenvironment (TME) that is pivotal in determining postsurgical treatment efficacy remains poorly understood. Here, with an experimental model, we demonstrate that the genomic landscape shaped by SR creates an immunosuppressive milieu characterized by hypoxia and high-influx of myeloid cells, fostering cancer progression and hindering PD-L1 blockade therapy. To address this issue, we engineer a radio-immunostimulant nanomedicine (IPI549@HMP) capable of targeting myeloid cells, and catalyzing endogenous $H_2O_2$ into $O_2$ to achieve hypoxia-relieved radiotherapy (RT). The enhanced RT-mediated immunogenic effect results in postsurgical TME reprogramming and increased susceptibility to anti-PD-L1 therapy, which can suppress/eradicate locally residual and distant tumors, and elicits strong immune memory effects to resist tumor rechallenge. Our radioimmunotherapy points to a simple and effective therapeutic intervention against postsurgical cancer recurrence and metastasis.

[1] Department of Medical Ultrasound and Center of Minimally Invasive Treatment for Tumor, Shanghai Tenth People's Hospital, School of Medicine, Tongji University, Shanghai 200072, P.R. China. [2] Ultrasound Research and Education Institute, Clinical Research Center for Interventional Medicine, School of Medicine, Tongji University, Shanghai 200072, P.R. China. [3] Shanghai Engineering Research Center of Ultrasound Diagnosis and Treatment; National Clinical Research Center for Interventional Medicine, Shanghai 200072, P. R. China. [4] Materdicine Lab, School of Life Sciences, Shanghai University, Shanghai 200444, P. R. China. [5] Department of Ultrasound, Zhongshan Hospital, Fudan University, Shanghai 200032, P. R. China. [6] These authors contributed equally: Xin Guan, Liping Sun, Yuting Shen. ✉email: chenyuedu@shu.edu.cn; xuhuixiong2022@126.com; yuewen0902@tongji.edu.cn

Surgery represents the mainstream therapeutic modality in oncology. However, despite advances in surgical technique and instrumentation, up to 30–40% of those patients recur within 5 years, mainly owing to the local residual tumor cells in surgical margins[1,2]. The tumor recurrence after surgical resection (SR), which is strongly associated with worse clinical outcome and poor overall survival, still remains challenging[3]. Recently, cancer immunotherapy has sparked profound hope to fight against tumor[4], and various types of immunotherapeutic strategies have been extensively utilized in preclinical studies and even clinical trials to prevent tumor relapse and metastasis[5,6]. Unfortunately, the responses, especially after surgery, to date have been infrequent despite the generation of certain antigen-specific cytotoxic T lymphocytes (CTL)[3,5].

For decades, research into improving therapeutic outcomes after SR focused almost entirely on the tumor cell itself, while ignoring the potential biological interactions between residual tumor and the microenvironment in which it grows. Therefore, classical postsurgical immunotherapy regimens to a great extent failed to introduce the specific effects of SR on the tumor microenvironment (TME). Indeed, mounting evidences have emphasized that the TME influenced by surgical manipulation can not only exert direct effects on the cancer cell to promote its proliferation, motility and invasion but also suppress the activity of antitumor leukocytes (e.g., natural killer (NK) cells, CTL and dendritic cells (DCs))[3,7]. Therefore, the re-engineered biological processes within it, should likely be crucial in determining the success or failure of postsurgical treatment. The attempts to reveal and inhibit the underlying principles responsible for such SR-modulated pro-oncogenic and immunosuppressive effects are highly required[1,3,8].

New strategies for tumor immunotherapy and radiotherapy (RT) have been significantly developed in parallel[6,9]. RT stands out as an optimal partner for immunotherapy not only because of the established safety profiles, but also because radiobiology has profound immunostimulatory effects that can synergize with other immuno-oncology agents in systemic tumor control[10]. However, the efficacy of conventional RT regimens is usually limited by tumor hypoxia-associated radiation resistance[11]. Fortunately, many clinic trials and conceptive studies have demonstrated that restoring tumor oxygenation level could substantially improve the efficacy of external beam RT[9,12]. Hence, the exploitation of hypoxia-mediated modalities for advanced therapeutic activity is of great interest[9].

Since the rapid development of nanotechnology, functionalized nanomaterials could be effectively utilized as adjuvant "drugs" for the oxygen-dependent standard therapies in the battle against solid tumors[13]. Among them, $MnO_2$-based nanosystems ($MnO_2$ NS), a unique TME-responsive agent, have attracted substantial attention[13,14]. Emerging researches suggest that $MnO_2$ NS can trigger the decomposition of tumor-overexpressed $H_2O_2$ into oxygen within the TME, which would greatly relieve the tumor hypoxia status. Meanwhile, the decomposed water-soluble $Mn^{2+}$ ions could enhance the contrast of $T_1$-weighted magnetic resonance (MR) imaging and are also rapidly excreted by kidneys, thus avoiding the long-term in vivo toxicity concerns for metal oxide that are prevalent in many other metal-based nanostructures[13,14]. Moreover, hollow $MnO_2$ ($HMnO_2$) nanostructures with large cavities have proven to be excellent drug delivery systems for loading therapeutic agents, whose release can be precisely controlled by tuning the shell coatings or structures[15].

Here, we describe an experimental tumor model system that links SR and its subsequent biological response to the progression of residual tumor cells. Specifically, we observe that SR shapes a unique genomic landscape, creating a highly immunosuppressive milieu marked by hypoxia and the expanded myeloid cell populations, which subsequently promotes residual tumors outgrowth and enhances resistance to programmed death-ligand 1 (PD-L1) blockade therapy. Recent evidence has emphasized a prominent role for immunomodulatory of tumor outgrowth[3,16], leading us to focus on the myeloid cell-induced immunosuppression as a mechanism regulating the postoperative cancer progression. Based on the above analysis, we develop a PEGylated $HMnO_2$ (HMP)-bridged radioimmunotherapy nanoplatform loaded with a small molecular PI3-kinase γ (PI3kγ) inhibitor (IPI549) that could overcome immune tolerance by eliminating these immunosuppressive myeloid cells[17]. In this system, HMP nanoparticles would be dissociated under specific acidic TME to enable tumor-targeted therapeutics delivery and pH-triggered on-demand drug release. Meanwhile, HMP nanoshells exhibit excellent catalase activity to decompose endogenous $H_2O_2$ into $O_2$, so as to achieve hypoxia-relieved postoperative RT. The PI3Kγ inhibition synergizes with the enhanced RT-mediated immunogenic cell death (ICD) effect, resulting in reprogramming of the post-resection immunosuppressive TME into an immunogenic phenotype with concomitant increased susceptibility to immune checkpoint blockade (ICB) therapy. Our results indicate that IPI549@HMP-based RT plus PD-L1 blockade leads to significant inhibition of locally residual and distant metastatic tumors, as well as tumor rechallenge (Fig. 1). Here, we emphasize the promise of modulating immunosuppressive TME with smart nanosystems to improve the efficiency of existing postsurgical therapies. Our radioimmunotherapy strategy is demonstrated to be a logical choice for preventing postsurgical cancer recurrence and metastasis, and experimental results suggest the potential clinical translation of this specific, effective, and low toxicity approach upon tumor resection.

## Results

**SR-driven immunosuppression accelerates local tumor progression.** To determine the effects of SR on modulating tumor-igenesis, a postsurgical CT26 colon mouse model, in which SR was performed to partially remove the tumor, was introduced in this study. Concurrently, sham operation was performed on the contralateral flank of tumor-bearing mice to determine whether the normal tissue injury caused by surgery can also lead to rapid tumor progression (Fig. 2a). We found that compared with tumors in untreated and sham operation groups, SR-treated mice initially decreased the size of the treated tumors. However, the residual tumors eventually grew larger than the tumors in the other two groups (Fig. 2b, c and Supplementary Fig. 1). Meanwhile, no obvious difference was observed between the sham operation and untreated groups. These results together suggested that the presence of residual tumor following SR induced accelerated cancer progression.

To clarify the potential molecular mechanism of the causal link between SR and local tumor outgrowth, we performed transcriptomic analysis using isolated post-SR residual tumors and untreated tumors three days after surgery. A total of 3823 differentially expressed gene profiles were identified including 2684 upregulated genes and 1139 downregulated genes (Fig. 2d, e and Supplementary Fig. 2). Gene Ontology (GO) analysis revealed that the differentially expressed genes were significantly enriched in the biological process (BP) of immune-related functions and pathways, mainly including immune system process, defense response, immune response and regulation of response to stimulus, suggesting a strong correlation between SR and immune-related functions. Meanwhile, they were enriched in molecular functions (MF) associated with receptor-binding such as protein binding and signaling receptor binding (Fig. 2f).

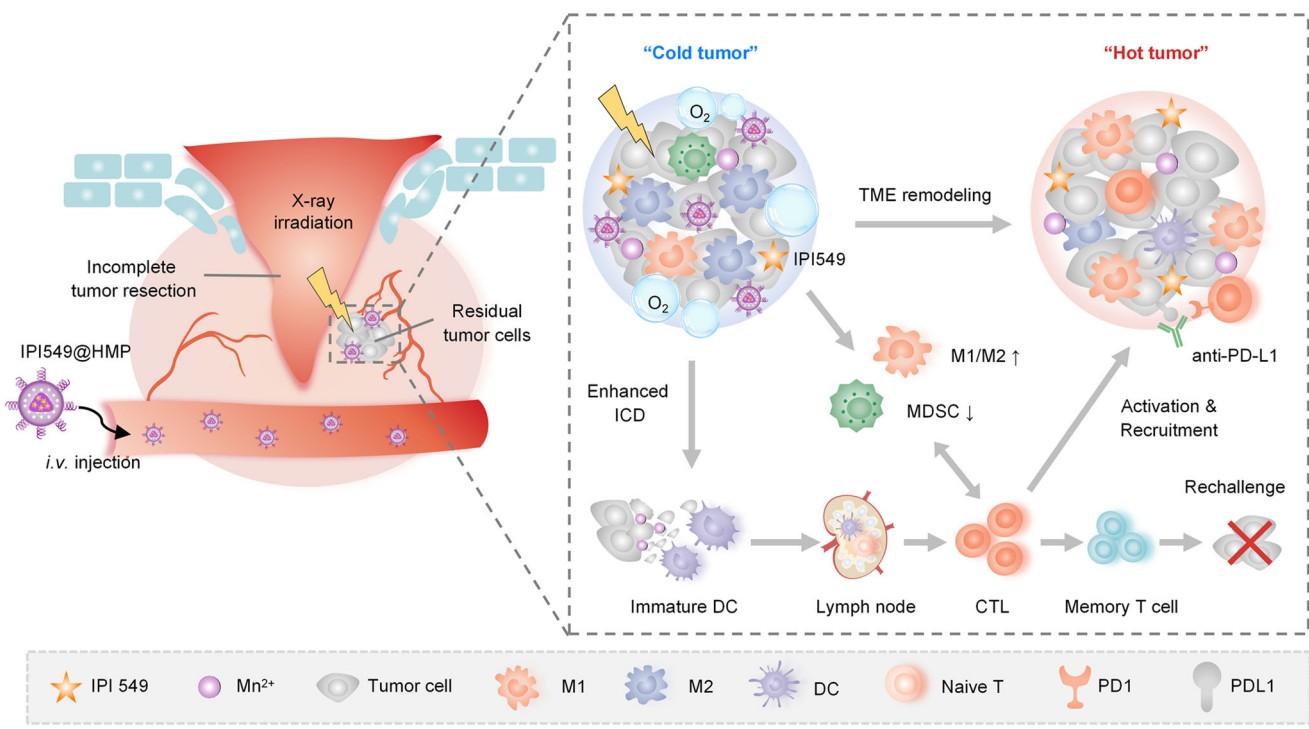

**Fig. 1 Schematic illustration of engineering radioimmunostimulant nanomedicine synergizing with PD-L1 blockade for postsurgical cancer immunotherapy.** Surgical resection (SR) creates an immunosuppressive milieu characterized by hypoxia and high-influx of myeloid cells. The radioimmunostimulant $HMnO_2$-based nanoplatform (IPI549@HMP) involved in this strategy enables hypoxia-relieved radiotherapy and pH-triggered release of IPI549 capable of targeting myeloid cells, which subsequently remodels immunosuppressive postresection TME into an immunostimulatory phenotype and heightens susceptibility to anti-PD-L1 therapy. IPI549@HMP-augmented radioimmunotherapy in combination with anti-PD-L1 leads to significant inhibition of locally residual and distant tumors, and elicits strong immune memory effect to completely resist tumor rechallenge. ICD, immunogenic cell death; M1, M1-like macrophage; M2, M2-like macrophage; MDSC, myeloid-derived suppressor cell; DC, dendritic cell; CTL, cytotoxic T lymphocyte.

Notably, genes encoding hypoxia, proinflammatory cytokines, chemokines, and genes related to immunosuppression were significantly up-regulated in the residual tumors (Fig. 2g). As expected, the over-presentation of genes in the Kyoto Encyclopedia of Genes and Genomes (KEGG) pathways was associated with tumor angiogenesis, inflammation and immunity (Fig. 2h). These data revealed that SR elicits a complex inflammatory reaction and hypoxia in the residual tumor, which ultimately promotes the development of immunosuppression in the postsurgical local-regional TME.

To further investigate the cellular mechanisms underlying SR-triggered immunoediting and selection, residual CT26 tumors were harvested and analyzed for immunologic phenotype changes by polychromatic immunofluorescent staining (PIF) and flow cytometry (FCM) three days after SR. PIF results displayed a significantly increased infiltration of CD45+ immune cells in SR-treated tumors compared to untreated one, and most of which were CD11b+ myeloid cells (Fig. 2i). Moreover, the proliferation and hypoxia indexes (displayed by Ki67 and HIF-α staining, respectively) were much higher within postsurgical tumor (Fig. 2i). Then, FCM analysis further confirmed that the proportion of CD45+ tumor-infiltrating leukocytes (TILs) in the SR group was 7.84 ± 1.68%, which was 3.4-fold higher than that of controls (2.32 ± 0.25%) (Supplementary Fig. 3a). As expected, myeloid-derived suppressor cells (MDSCs) constituted the majority of CD45+ TILs in this model (Fig. 2j, m). We then characterized the CD11b+ subpopulations including M1-like macrophages (TAMs-M1) and M2-like macrophages (TAMs-M2)[18]. As is known, TAMs-M1 perform proinflammatory functions and exhibit robust effector functions against pathogens

and cancer cells, while TAMs-M2 perform anti-inflammatory functions capable of promoting the outgrowth and metastasis of tumor cells[19]. Comparing to the untreated controls, macrophages infiltration polarized to the M2 phenotype were substantially increased in SR-treated tumors (Fig. 2k, n, o and Supplementary Fig. 3b). Simultaneously, we observed a reduced infiltration of CD8+ T cells and a decreased CD8+ T/CD11b+ cell ratio (Fig. 2l, p, q), which strongly correlated with the reported T cell suppression effects induced by tumor-associated myeloid cells (TAMCs)[20]. Additionally, the proportion of Treg (CD4+Foxp3+) was unaffected, further confirming that SR-driven immunosuppression was mainly derived from the high influx of myeloid cells (Supplementary Fig. 3c). Taken together, these results demonstrate that residual tumors post-SR harbour an aggravated immunosuppressive and exhaustive TME characterized by enhanced accumulation of myeloid cells with a paucity of CTL infiltration, and such a gradient towards immunologic inactivation indicates the TME reconstructing and immune evolution, which have important implications in the promoted tumor progression.

**PI3Kγ as the research target for postsurgical treatment.** To efficiently eliminate myeloid cell-associated immunosuppression and reactivate the anticancer responses after SR, PI3Kγ, a key molecular switch associated with multiple immune-related signaling pathways, serves as a viable target molecule for postsurgical treatment[17]. PI3Kγ protein is the most highly expressed PI3K-isoform on MDSCs, and it functions downstream of diverse chemoattractant–receptor pairs to promote the recruitment of

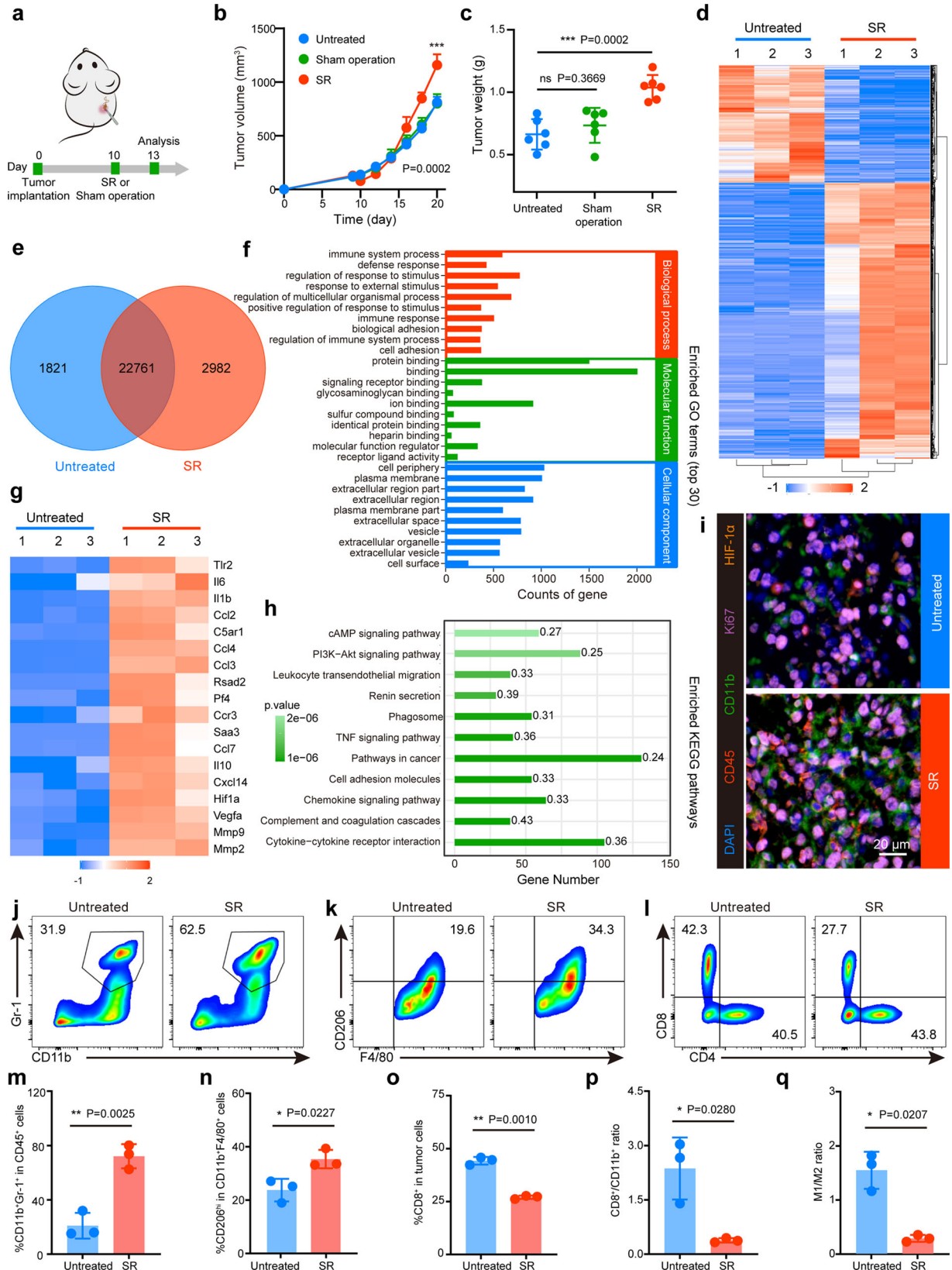

MDSCs into the tumor[17]. Correspondingly, blocking the PI3Kγ-mediated pathways may exert a repolarizing effect on TAMs and activated T cells[17,21]. Therefore, this process plays an vital role in forming immunosuppressive TME and limiting the effective anticancer immunity. Interestingly, we observed an obviously upregulation of PI3Kγ protein expression in residual tumor tissues compared with controls using immunofluorescence staining (Supplementary Fig. 4). Thus, we propose to use IPI549, a selective PI3Kγ inhibitor confirmed in multiple tumor models[22,23] to reverse myeloid cell-mediated immunosuppression, which in combination with RT that can cause ICD effect to trigger certain levels of antitumor immunity owing to the

**Fig. 2 SR-driven immunosuppression accelerates local tumor progression. a** Schematic illustration of surgical resection (SR) treatment. $1\times10^6$ CT26 cells were subcutaneously injected into the right flank of BALB/c mice. SR or sham operation was conducted on the right tumor or left skin, respectively, on day 10 postinoculation. **b** Residual tumor growth kinetics of mice in Untreated, SR and Sham operation groups ($n = 6$ mice). **c** Weight of the excised tumor on day 20 after varied treatments ($n = 6$ mice). **d, e** Cluster analysis (**d**) and Venn diagram (**e**) of differential expression genes in RNAseq between untreated and SR-treated tumors three days post treatment ($n = 3$ mice). Red and blue colors represent upregulated or downregulated genes, respectively. **f** Significant enrichment in gene ontology (GO) terms (top 30, $n = 3$ mice). **g** Heatmap of differentially expressed genes associated with tumor progression and immunosuppression ($n = 3$ mice). Red and blue colors represent upregulated or downregulated genes, respectively. **h** Kyoto Encyclopedia of Genes and Genomes (KEGG) enrichment histogram of differentially expressed genes (Statistical difference was calculated using Fisher's exact test, $n = 3$ mice). **i** Representative polychromatic immunofluorescent staining images of tumors from three biologically independent samples showing CD45$^+$ (red), CD11b$^+$ (green), Ki67$^+$ (purple) and HIF-1α$^+$ (orange) cells infiltration for Untreated and SR tumors three days post-treatment. **j–o** Representative flow cytometric images and the corresponding quantification of MDSCs (CD11b$^+$Gr-1$^+$CD45$^+$) (**j**, **m**), TAMs-M2 (CD206$^{hi}$CD11b$^+$F4/80$^+$CD45$^+$) (**k**, **n**) and CTLs (CD8$^+$CD3$^+$CD45$^+$) (**l**, **o**). **p** The ratio of CD8$^+$ cells to CD11b$^+$ MDSCs in tumors. **q** The ratio of M1 to M2 in tumors. SR, surgical resection; MDSCs, myeloid-derived suppressor cells; TAMs-M2, M2-like macrophages; TAMs-M1, M1-like macrophages; CTLs, cytotoxic T lymphocytes. Data were expressed as means ± SD ($n = 3$ biologically independent samples). Statistical difference was calculated using two-tailed unpaired student's $t$-test. ns, not significant, $*P < 0.05$, $**P < 0.01$ and $***P < 0.001$. The experiments were repeated three times. Source data are provided as a Source Data file.

exposure of tumor-associated antigens in necrotic cell debris, should likely stimulate a potently enhanced radioimmunotherapy effect to finally destroy the incompletely removed tumor cells.

**Synthesis and characterization of HMP-based nanosystem.** Next, we designed an HMP-based nanosystem for tumor-targeted IPI549 delivery and TME-triggered release to target MDSCs. The IPI549@HMP nanosystem was synthesized as illustrated in Fig. 3a. Briefly, monodisperse solid silica spheres (sSiO$_2$) were firstly fabricated by a classical stöber method[24]. After reacting with potassium permanganate (KMnO$_4$) solution, the surface of sSiO$_2$ was coated with uniform MnO$_2$ layer. Then, the obtained sSiO$_2$@MnO$_2$ was etched with Na$_2$CO$_3$ solution to obtain HMnO$_2$. After that, the obtained HMnO$_2$ nanoparticles were sequentially modified and drug loaded to yield the final IPI549@HMP nanosystem for further experiments.

Transmission electron microscope (TEM) images clearly displayed the spherical morphology and hollow structure of HMP (Fig. 3b). The chemical status of the HMP nanocomposites were analyzed using X-ray photoelectron spectroscopy (XPS), and the relative contents of Mn$^{2+}$, Mn$^{3+}$, and Mn$^{4+}$ were determined to be 56.9, 17.0, and 25.9%, respectively (Supplementary Fig. 5). Element mapping analysis further confirmed the existence and uniform distribution of Mn and O elements in the HMP nanostructure (Fig. 3c). Next, N$_2$ absorption-desorption measurement was performed to estimate the permanent porosity of HMnO$_2$ nanoparticles, and its surface area and average pore diameter were determined to be 349.5 m$^2$g$^{-1}$ and 4.1 nm, respectively, which is expected to be a desirable nanocarrier for efficient drug loading (Fig. 3d). The successful construction of HMP and the encapsulation of IPI549 was proved by the shifting zeta potential during the stepwise synthesis and also validated by the characteristic absorption peak of IPI549 at ~249 nm in the UV-Vis absorption spectrum of IPI549@HMP (Fig. 3e, f). The average particle sizes of HMP and IPI549@HMP nanoparticles were around 143 nm and 162 nm, respectively, as determined by the dynamic light scattering technique (Fig. 3g). Notably, IPI549@HMP exhibited well stability in different physiological media without obvious aggregation. The physical properties of IPI549@HMP including the nanoscale size and high dispersity benefit the following in vitro performance evaluation and in vivo biomedical use.

MnO$_2$ is known to be stable under neutral pH, but could be decomposed into Mn$^{2+}$ ions under TME conditions with low pH and high H$_2$O$_2$ contents[25]. Therefore, the degradation behaviors of HMP nanoparticles in a mimicked TME were first evaluated (Supplementary Fig. 6). As expected, HMP nanoparticles showed no significant change in neutral environment (pH 7.4) even after

24 h, while they exhibited time-dependent degradation behavior in an acidic environment (pH 6.5) mainly attributing to the decomposition of MnO$_2$ into Mn$^{2+}$ ions. Remarkably, the release amount of Mn$^{2+}$ increased dramatically after the addition of H$_2$O$_2$ and the degradation behavior was positively correlated with the H$_2$O$_2$ concentration. Besides, by adjusting the initial feeding ratio of HMP and IPI549, the optimal drug-loading rate was determined to be about 69.8% (Supplementary Fig. 7). Consistent with the degradation behavior of HMP itself, the cumulative release kinetics of IPI549@HMP manifested faster and more drug release in acidic solutions, especially containing H$_2$O$_2$ (Fig. 3h). These data demonstrated the ultrasensitive TME-responsive drug release efficiency of IPI549@HMP.

Early studies have suggested that Mn$^{2+}$ can enhance T$_1$-weighted MR imaging by promoting proton transfer and prolonging longitudinal relaxation[16]. Thus, the potential T$_1$-weighted MR imaging capability of IPI549@HMP was investigated. The concentration-dependent brightening effects of IPI549@HMP nanoparticles were obviously monitored in T$_1$-MR images at pH 6.5, whereas its signals in neutral solution (pH 7.4) appeared to be much weaker (Fig. 3i). Specially, the r$_1$ value improved significantly from the initial value of 0.44 mM$^{-1}$s$^{-1}$ at pH 7.4 to 7.10 mM$^{-1}$s$^{-1}$ after incubation in pH 6.5 containing H$_2$O$_2$ (100 μM) buffer (Fig. 3j and Supplementary Fig. 8). Next, we investigated the biosafety of HMP as a multifunctional nanocarrier in vitro and in vivo. Cytotoxicity assays indicated negligible toxicity of IPI549@HMP to CT26 and B16F10 cancer cells, even at doses up to 100 ppm (Supplementary Fig. 9). Similarly, the blood cell hemolysis rate was still lower than 5% even at HMP concentration of 200 ppm (Fig. 3k). Furthermore, serum biochemical assays showed no significant variations and the hematoxylin and eosin (H&E) staining of the major organs also exhibited no obvious pathological abnormalities compared with the normal mice, indicating the excellent histocompatibility of IPI549@HMP nanoparticles (Supplementary Fig. 10). Taken together, these results proved that IPI549@HMP nanoparticles with low toxicity can be safely administered intravenously. Then, the in vivo tumor-specific MR imaging performance of IPI549@HMP nanoparticles was further evaluated. It was found that the T$_1$-MR signal in the tumor area was remarkably enhanced after the injection (Fig. 3l). These data confirmed the high biocompatibility of IPI549@HMP and its excellent tumor-specific MR imaging capability.

**In vivo biodistribution and pharmacokinetics of IPI549@HMP.** The distribution and tumor accumulation of IPI549@HMP in CT26 tumor-bearing mice were evaluated by tracking the fluorescence of ICG-labeled IPI549@HMP using an IVIS spectrum imaging system. As shown in Supplementary Fig. S11a, b, the ICG fluorescence

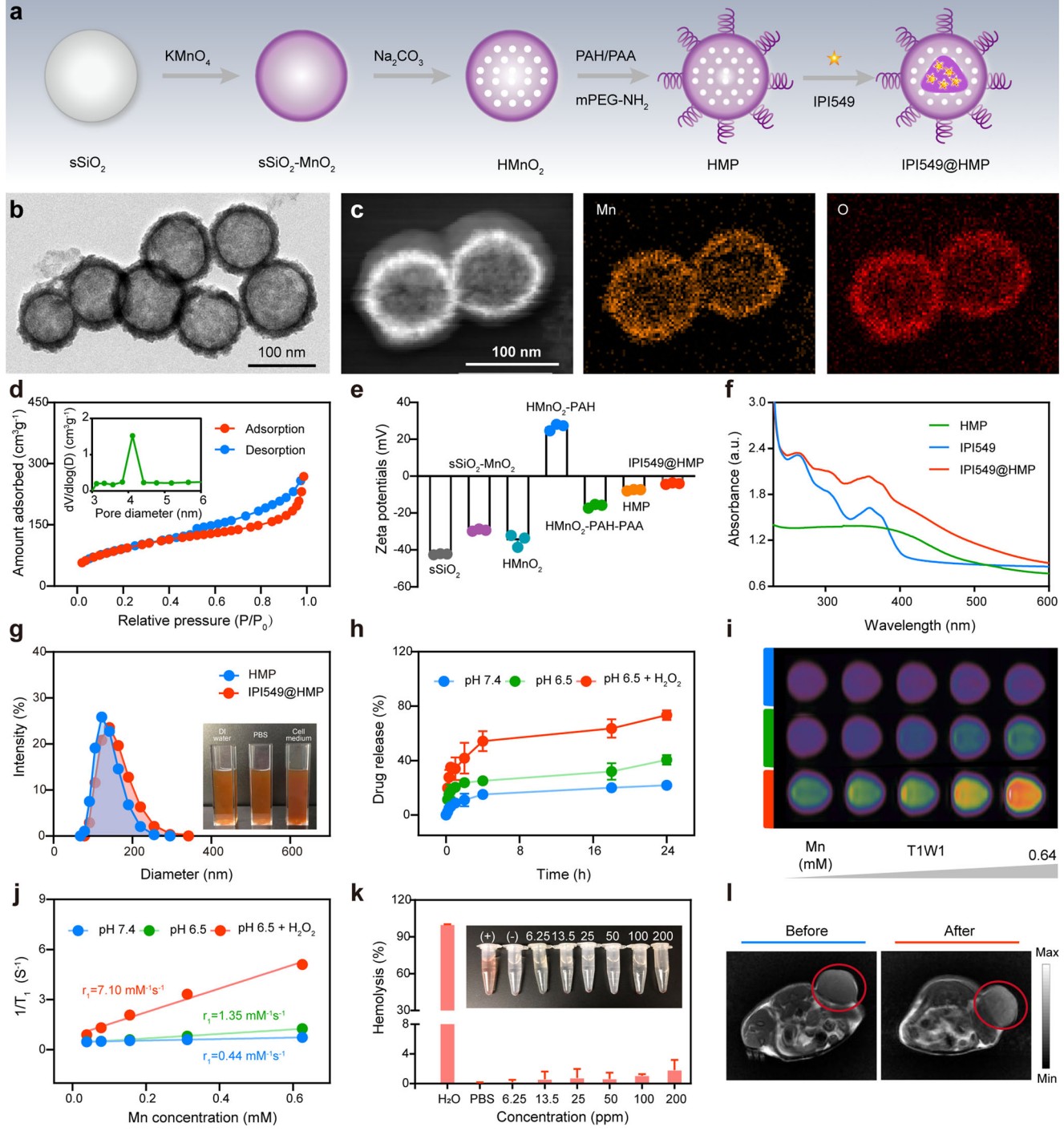

**Fig. 3 Schematic and characterization of HMP-based nanoplatform. a** A scheme indicating the step-by-step synthesis of HMP nanoparticles and the subsequent drug loading. **b** The representative TEM image of HMP nanoparticles from three independent samples. **c** The representative HAADF-STEM image and corresponding elemental mappings of HMP nanoparticles from three independent samples. **d** The representative $N_2$ absorption-desorption isotherms and pore size distribution (inset) of $HMnO_2$ nanoparticles from three independent samples. **e** Zeta potential variations in the preparation procedure of IPI549@HMP nanoparticles. Data were expressed as means ± SD ($n = 3$ independent samples). **f** Representative UV-vis spectrums of free IPI549, HMP and IPI549@HMP from three independent samples. **g** Representative particle-size distributions of HMP and IPI549@HMP from three independent samples (Inset: digital photos of IPI549@HMP dispersed in deionized water, PBS and cell culture medium). **h** Cumulative release kinetics of IPI549 from HMP in varied conditions. **i** Representative $T_1$-weighted MR images of different concentrations of IPI549@HMP dispersed in varied conditions from three independent samples. **j** Representative relaxation rate $r_1$ *versus* $Mn^{2+}$ concentrations when dispersed in varied conditions from three independent samples. **k** Concentration-dependent hemolysis and relative digital photo (inset) of IPI549@HMP. Data were expressed as means ± SD ($n = 3$ independent samples). $H_2O$ and PBS were set as positive and negative control, respectively. **l** Representative $T_1$-weighted MR images of CT26 tumor-bearing mice before and after IPI549@HMP intravenous injection from three biologically independent samples. The red circle indicates tumor tissue. Source data are provided as a Source Data file.

intensity in the tumor area increased with time and reached at a peak level at 8 h after injection. Semi-quantitative biodistribution according to isolated of major organs indicated high tumor uptake and retention of IPI549@HMP (Supplementary Fig. S11c, d). Notably, distinct fluorescence in the liver and kidneys was presented as expected, since the platform starts to degrade and be excreted over this time period. Meanwhile, the blood-circulation half-time was calculated to be 0.97 h (Supplementary Fig. S11e), which indicates the easy elimination of the IPI549@HMP from the central chamber, such as the kidney and liver.

**In vitro detection of IPI549@HMP-augmented RT**. The hypoxic TME of solid tumors is responsible for the limited efficacy of oxygen-dependent RT[26,27]. The MnOx-based nanosystems have been well proved to trigger the decomposition of endogenous $H_2O_2$ within TME to generate $O_2$, thereby improving in situ oxygen levels to enhance oxygen-dependent therapies[28] (Fig. 4a). Thus, we initially tested the catalytic oxygen production capacity of the designed IPI549@HMP in vitro. Rapid production of oxygen bubbles was observed in $H_2O_2$ solution (100 μM) using two-dimensional ultrasound in the presence of IPI549@HMP (Fig. 4b). Consistently, dynamic monitoring of dissolved oxygen changes with an oxygen probe further verified that oxygen production was proportional to the Mn concentrations (Fig. 4c). Moreover, the decreased expression of HIF-1α at the cellular level also confirmed the excellent hypoxia relief efficacy of IPI549@HMP (Supplementary Fig. 12).

Double strand breaks (DDSB) that has been well-recognized as the most lethal type of damage induced by the ionizing radiation, is a major indicator of RT efficacy[29]. Then, the effect of HMP on enhancing radiation response was evaluated by using a specific marker-γ-H2AX staining. As expected, we observed that when incubated with HMP or IPI549@HMP and irradiated with X-rays, significant γ-H2AX fluorescence was observed in the nuclei of CT26 cells, indicating an increased DNA damage of the combined treatment (Fig. 4d). Consistently, calcein-AM/PI staining of CT26 multicellular spheroids showed that treatment with HMP/IPI549@HMP appeared to be much more effective in inhibiting the proliferation of X-ray-irradiated cells than RT alone, likely due to the additional oxygen supplied by $HMnO_2$-triggered $H_2O_2$ decomposition within the tumor cells (Fig. 4e). Meanwhile, the long-term radiosensitizing effect of IPI549@HMP was estimated via colony formation assay, which displayed that IPI549@HMP plus RT restrained colony formation to a greater extent than RT alone (Supplementary Fig. 13). Therefore, IPI549@HMP is expected to be an effective radiosensitization agent under hypoxic TME.

**IPI549@HMP-augmented RT against postsurgical tumor progression**. Encouraged by the superior performance of HMP-based combination therapy in vitro, we then investigated its therapeutic efficacy in animal tumor models. Photoacoustic (PA) imaging was firstly utilized to assess the tumor-targeting ability and hypoxia-relief capacity of IPI549@HMP[30]. The PA signal intensity of HMP in vitro was positively correlated with its concentrations (Supplementary Fig. 14). Subsequent intravenously injected with IPI549@HMP into CT26 tumor-bearing mice displayed a gradual increase of PA signal in the tumor region and peaked at 8 h postinjection, indicating a high accumulation of these nanoparticles in the tumor region (Fig. 4f, g). Meanwhile, the variation of vascular saturated $O_2$ ($sO_2$) levels detected using oxyhemoglobin/deoxyhemoglobin mode indicated that tumors in mice treated with IPI549@HMP exhibited dramatically increased $sO_2$ levels. Additionally, HIF-1α staining confirmed that HMP-based treatment indeed alleviated the tumor

hypoxic state (Fig. 4h, i). Together, these results demonstrated the excellent tumor enrichment and hypoxia-relieving capacity of IPI549@HMP.

Next, the efficacy of IPI549@HMP augmented RT to prevent colon cancer relapse after SR was evaluated. On day 10 after inoculation with Luc+ CT26 cells, the visible tumor was partially surgically removed to establish a postsurgical tumor model (Fig. 5a). Then, the BALB/c mice bearing residual tumors were randomly divided into five groups ($n = 6$) before each treatment, including Control group (PBS), IPI549@HMP group, RT group, HMP + RT group, IPI549@HMP + RT group (dose of $MnO_2 = 7.5$ mg kg$^{-1}$ and IPI549 = 1.5 mg kg$^{-1}$). The mice were intravenously injected with PBS/HMP/IPI549@HMP according to groups on the 1st and 3rd post-operation, followed by two successive X-ray irradiation (3 Gy) on 8 h post-injection for two cycles. Bioluminescence images showed no appreciable suppressive effect of free IPI549@HMP on tumor growth (Fig. 5b–e and supplementary Fig. 15). Although RT alone caused a partial delay in tumor growth, it exhibited stronger tumor growth-inhibition effect when combined with HMP. Of note, after combined IPI549@HMP with X-ray irradiation treatment, the residual tumors exhibited the slowest outgrowth speed and all the mice survived for 60 days, with complete responses in 50%. Meanwhile, although the weight of mice undergoing RT treatment initially decreased slightly due to frequent anesthesia, there was no evident difference in the subsequent days compared to the controls (Fig. 5f). Serum biochemistry assay and the histology analysis of major organs obtained from mice 7 days after IPI549@HMP–based RT treatment indicated no noticeable abnormality (Supplementary Fig. 16).

To further explore whether such a combined radioimmunotherapy strategy could potentially be used for other tumor types, we employed murine B16F10 melanoma as the model in addition to the colon CT26 cancer model. Similar to previous observations, IPI549@HMP + RT also exhibited remarkable therapeutic efficacy against murine postsurgical residual melanoma and significantly reduced the tumor growth rate compared to controls (Supplementary Fig. 17). These results successfully proved that our combined radioimmunotherapy strategy in CT26 cancer model can be extended to other types of tumors.

**Immunological responses after IPI549@HMP-augmented RT**. Considering the superior capability of the combined radioimmunotherapy for postoperative tumor control, we wondered whether IPI549@HMP-augmented RT would have certain effect on the aggravated immunosuppressive phenotype of residual tumors. Western blot assay of representative tumors indicated that the IPI549-loaded nanoparticles did inhibit the expression levels of PI3Kγ, as also verified in immunofluorescence staining (Fig. 5g and Supplementary Fig. 18). Meanwhile, significantly abundant expression of calreticulin (CRT) and high mobility group box 1 (HMGB1) was detected after IPI549@HMP-based RT treatment compared to the controls (Fig. 5h). These results suggested that the optimal inhibitory efficiency of IPI549@HMP plus RT owing to the synergistic effects of enhanced ICD effect and reversal of immunosuppression.

Accordingly, we investigated how intratumoral immune cells mediate the therapeutic effects of the HMP-enabled RT process when PI3Kγ-related pathways were systemically inhibited. Changes in leukocyte phenotypes within the TME were systematically evaluated by FCM and PIF after different kinds of treatments. Localized radiation initiates cell death and the production and release of cytokines as well as chemokines into the TME, which leads to infiltration of tumor-associated DCs (Supplementary Fig. 19). FCM analysis revealed that IPI549@HMP-based RT effectively enhanced

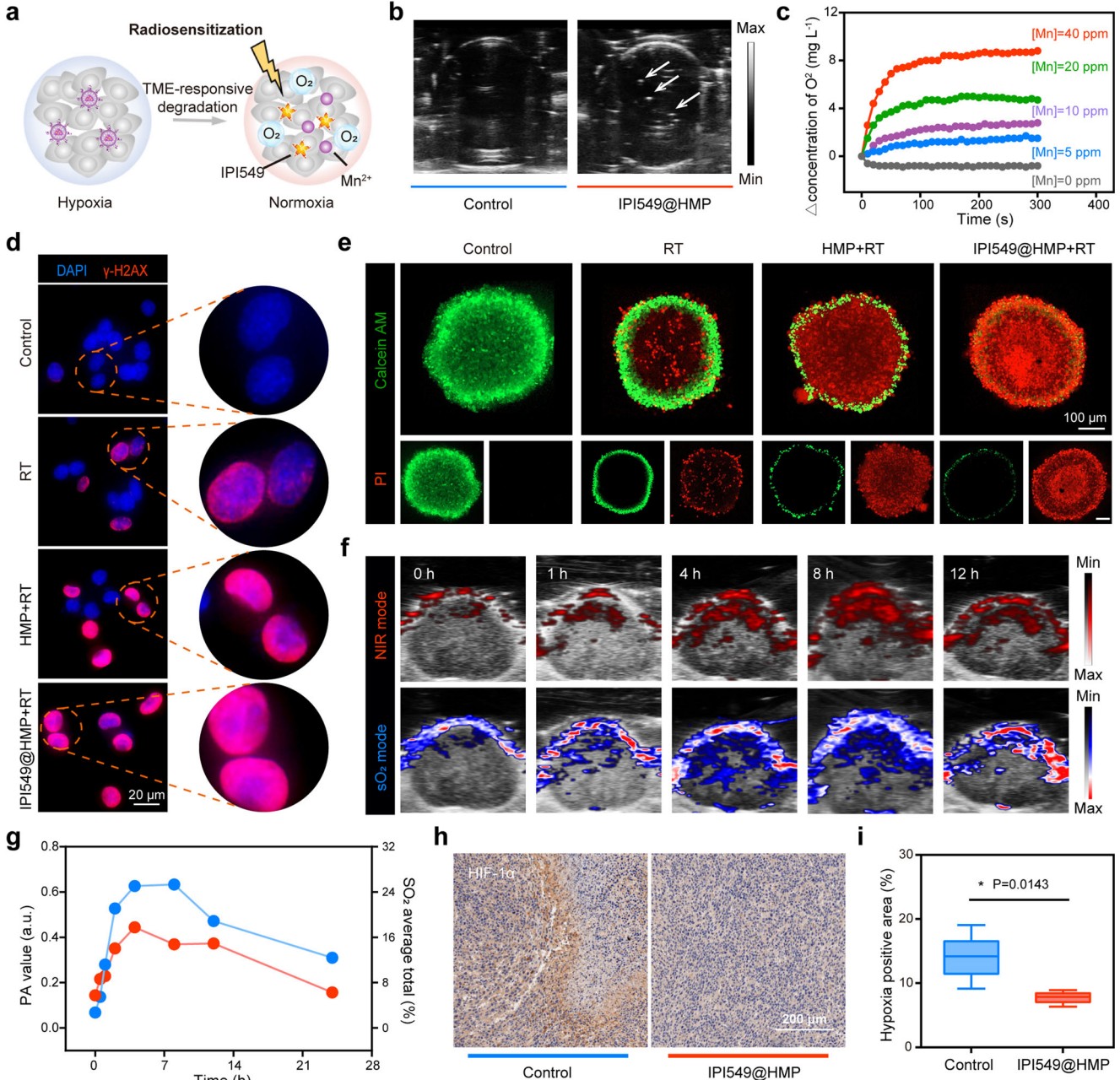

**Fig. 4 IPI549@HMP-induced hypoxia remission for augmented RT. a** Schematic illustration of IPI549@HMP for hypoxia-relieved RT. **b** Representative ultrasonic image of IPI549@HMP reacted with $H_2O_2$ (100 μM) of three independent samples from each group. The white arrow points to the generated oxygen bubble. **c** Time-dependent oxygen production at varied Mn concentrations as detected by a portable dissolved oxygen analyzer ($n = 3$ independent samples). **d** Representative DNA damage marker γ-H2AX assays of CT26 tumor cells after varied treatments of three biologically independent samples from each group. **e** Representative confocal images of CT26 multicellular spheroids (MCSs) treated with varied samples and stained with Calcein AM (green) and PI (red) from three biologically independent samples. **f, g** Photoacoustic images (**f**) and corresponding quantification (**g**) in near-infrared mode (upper) and oxygen saturation mode (below) of tumors at varied time points post IPI549@HMP injection ($n = 3$ mice). Blue color indicates IPI549@HMP enrichment while red color represents oxygen saturation. **h, i** Representative HIF-1α immunohistochemical staining of three biologically independent samples from each group (**h**) and relative quantitative analysis (**i**) of tumor sections after *i.v.* injection of IPI549@HMP. The bold lines, upper boundaries and lower boundaries of notches represent the mean, max and min values. Data were expressed as means ± SD ($n = 5$ images per group). Statistical difference was calculated using two-tailed unpaired student's *t*-test. *$P < 0.05$. Source data are provided as a Source Data file.

the frequency of tumor-infiltrating CD8$^+$ T cells and reduced MDSCs and TAMs-M2 compared with other controls (Fig. 6a–f and Supplementary Fig. 20). Further subdivision of these myeloid cells into neutrophils MDSC (CD11b$^+$Ly6G$^{high}$) and monocytes MDSC (CD11b$^+$Ly6C$^{high}$) revealed that IPI549@HMP mainly impaired neutrophils MDSC recruitment, while the combination treatment reduced both neutrophils MDSC and monocytes MDSC infiltration

by 57 and 53%, respectively (Supplementary Fig. 21). These results suggested our IPI549@HMP plus RT strategy can efficiently hinder MDSC recruitment. Notably, the ratios of CD8/Treg, CD8/MDSC, and M1/M2, well-recognized as indicators of antitumor immune homeostasis, were highly improved in the IPI549@HMP + RT group (Fig. 6g, h and Supplementary Fig. 22). These results were further confirmed by the PIF analysis, which yielded consistent results

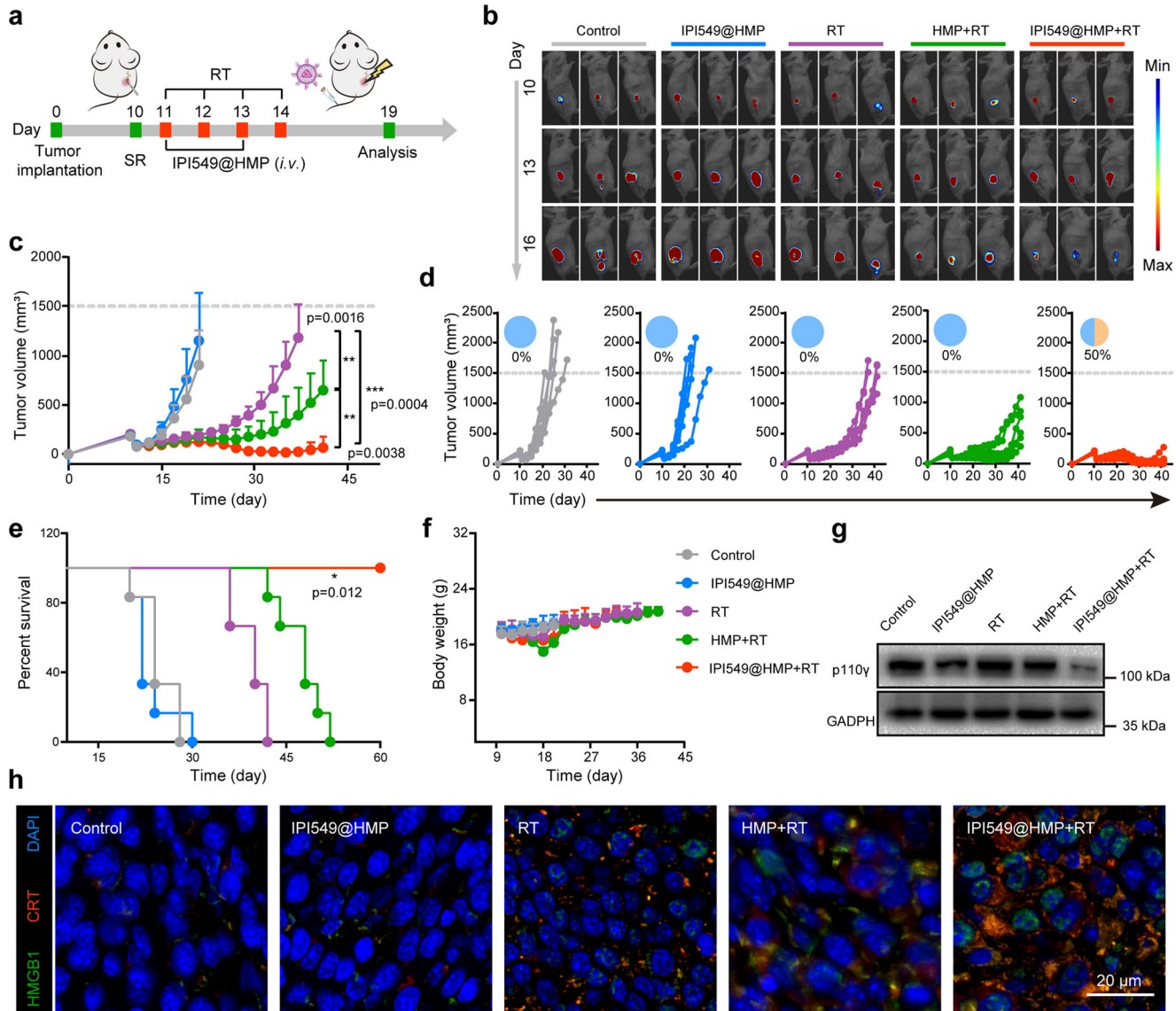

**Fig. 5 IPI549@HMP-augmented RT against postsurgical tumor progression. a** Schematic illustration of the experiment design to assess the in vivo IPI549@HMP-based RT and its triggered immune responses. **b** Representative bioluminescence images of Luc+ CT26 tumor after varied treatments as indicated (*n* = 6 mice). **c-f** Average tumor growth curves (**c**), individual tumor growth kinetics (**d**), Kaplan-Meier survival curves (**e**) and body weight fluctuation curves (**f**) of CT26 tumor-bearing mice after varied therapeutic combinations (*n* = 6 mice in **c-f**). **g** Western blot analysis of p110γ in residual tumors collected from mice in different groups. The experiments were repeated three times. **h** Representative immunofluorescence images from three biologically independent samples of tumor slices stained with DAPI (blue), CRT (red) and HMGB1 (green) antibodies. SR, surgical resection; RT, radiotherapy; CRT, calreticulin; HMGB1, high mobility group box 1. Statistical difference was calculated using two-tailed unpaired student's *t*-test (**c**) and Log-rank (Mantel-Cox) test (**e**). Data were expressed as means ± SD (**c**, **f**). *$P < 0.05$, **$P < 0.01$ and ***$P < 0.001$. Source data are provided as a Source Data file.

(Fig. 6i). Moreover, IPI549@HMP + RT-treated mice secreted the highest tumor necrosis factor-α (TNF-α) and interferon-γ (IFN-γ) in serum, which again verified the strong immune responses triggered by such radioimmunotherapy strategy (Fig. 6j, k). These data suggest that IPI549@HMP augmented RT not only increases the proportions of positive immune responders within postsurgical CT26 residual tumors, but also suppresses those negative immune inhibitors, thus eventually establishing an inflamed tumor immunity niche that exerts effective tumoricidal immune activity and inhibits postsurgical cancer growth.

To exclude the effect of potential immunogenicity of luciferase antigens, we repeated the tumor surgical resection study using wild type CT26 cells without luciferase expression. As shown in Supplementary Fig. 23, a similarly beneficial effect of the combination therapy (IPI549@HMP plus RT) in suppressing tumor recurrence was observed. In order to further confirm which T cell subtype plays a key role in tumor control, we performed CD8+ T cell and CD4+ T cell depletion experiments on a postsurgical colon cancer mouse model. It has been observed that IPI549@HMP + RT treatment lost most of the immunotherapeutic effect in primary CT26 tumors after CD8+ T cells depletion (Supplementary Fig. 23). However, the combined radioimmunotherapy still greatly inhibited tumor growth after CD4+ T cells depletion. These results indicated that CD8+ T cells deeply involved in IPI549@HMP mediated radiation sensitization and immunotherapeutics. Meanwhile, we found a significantly increment of NK cells infiltration within the tumor site after combination therapy by NKp46 immunofluorescence staining

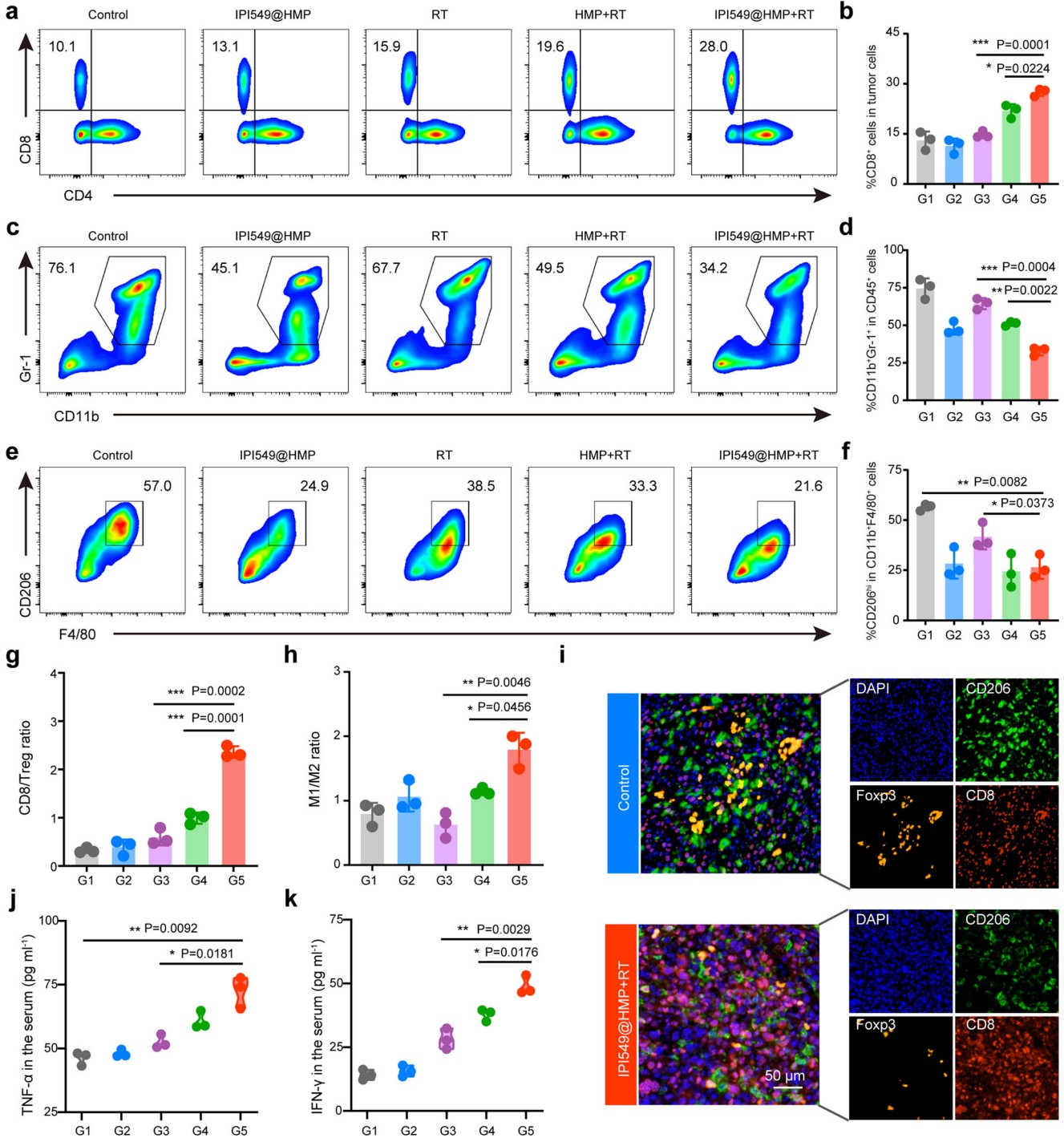

**Fig. 6 Robust antitumor immune responses triggered by IPI549@HMP-augmented RT. a–f** Representative flow cytometric analysis and relative quantification of CTLs (CD8+CD3+CD45+) (**a**, **b**), MDSCs (CD11b+Gr-1+CD45+) (**c**, **d**) and TAM-M2 (CD206hiCD11b+F4/80+CD45+) (**e**, **f**). **g**, **h** Quantification by flow cytometry of CD8/Treg (**g**) and M1/M2 ratios (**h**). **i** Representative polychromatic immunofluorescent staining images of tumors from three biologically independent samples showing DAPI (blue), CD206+ (green), Foxp3+ (orange) and CD8+ (red) cells infiltration for Control and IPI549@HMP + RT groups. **j**, **k** Cytokine levels of TNF-α (**j**) and IFN-γ (**k**) in the serum after varied treatments. G1, Control; G2, IPI549@HMP; G3, RT; G4, HMP + RT; G5, IPI549@HMP + RT. RT, radiotherapy; CTLs, cytotoxic T lymphocytes; Tregs, regulatory T cells; MDSCs, myeloid-derived suppressor cells; TAMs-M2, M2-like macrophages; TAMs-M1, M1-like macrophages. Data were expressed as means ± SD ($n = 3$ biologically independent samples in **b**, **d**, **f**, **g**, **h**, **j**, and **k**). Statistical difference was calculated using two-tailed unpaired student's $t$-test. *$P < 0.05$, **$P < 0.01$, ***$P < 0.001$. Source data are provided as a Source Data file.

(Supplementary Fig. 24), indicating that IPI549@HMP + RT could augment both innate and adaptive immune responses against tumors, thereby decreasing immunosuppression and potentiating the responsiveness of tumors to radiation.

**Abscopal effect of RT plus PD-L1 blockade**. Since the majority of cancer-induced deaths after surgery are caused by tumor metastases, ideal postsurgical treatment should not only locally inhibit the residual tumor but also recognize, suppress and remove the metastases. Recent years have witnessed the wide blooming of immunotherapy based on ICB, where PD-1/PD-L1 checkpoint blockade in combination with other treatment protocols, including RT, has attracted a great deal of research interest[31,32]. Given that genes highly correlated with clinical ICB resistance[33] were over-expressed among postsurgical tumor tissues (Supplementary Fig. 25), we wondered whether IPI549@HMP-augmented RT could further enhance sensitivity to PD-L1 blockade. Purposefully, a bilateral CT26 tumor-bearing mice model was employed. Five days after the primary tumor inoculation (on the right flank), a second tumor was implanted on the left flank to mimic the distant tumor (Fig. 7a). The right primary tumors were treated as before, in addition to *i.p.* injection of anti-PD-L1 (dose = 3.75 mg kg⁻¹) for PD-L1 blockade on day 11 and 13[6,34]. Treatment results revealed that anti-PD-L1 alone indeed exhibited no obvious effects on the inhibition of both primary and distant tumors (Fig. 7b–e and Supplementary Fig. 26). Although IPI549@HMP-augmented RT alone could somewhat suppress the primary tumor growth, it failed to exert influences on the mimic distant tumor. Notably, IPI549@HMP-augmented RT combined with PD-L1 blockade not only eradicated the primary residual CT26 tumor, but also remarkably suppressed the distant tumor growth. Moreover, no abnormal weight loss of mice was observed in the IPI549@HMP + RT plus anti-PD-L1 group (Supplementary Fig. 27).

To elucidate the therapeutic mechanism of radioimmunotherapy combined with PD-L1 blockade, immune cells in distant tumors were assessed on 19 days after primary tumor inoculation. The FCM results displayed a remarkably increase in the number of intratumoral CTL of the combination treatment group (Fig. 7f, i). Concomitantly, TAMs-M2 and MDSCs in distant tumors without RT treatment were substantially decreased compared to other controls including anti-PD-L1 alone and RT with IPI549@HMP (Fig. 7g–k). Notably, the antitumor immune balance indicators (*e.g.*, CD8/Treg and M1/M2) were obviously improved in the combination therapy group (Fig. 7l, m and Supplementary Fig. 28). In support, PIF also confirmed the above results (Fig. 7n). Besides, the Ki67 expression was remarkably reduced in the IPI549@HMP-augmented RT plus aPDL1 group, confirming that this combination therapy effectively impeded the tumor proliferation. Additionally, serum cytokine assays including TNF-α and IFN-γ showed that IPI549@HMP + RT plus aPD-L1 induced the highest levels of cytokine secretion in comparison to all other controls (Fig. 7o, p). Furthermore, PD-L1 expression level was found to be increased in both primary and distant tumors after IPI549@HMP + RT treatment (Supplementary Fig. 29). This upregulated PD-L1 expression should provide an opportunity for PD-L1 blockade that would uncover the full cytotoxic potential of host immunity against tumors. To reveal the potential biosafety of this intensified immunotherapy, serum biochemistry assay and pathological analysis of major organs were conducted. Results exhibited no obviously differences in either serum indexes or major organs between the IPI549@HMP-augmented RT plus aPDL1-treated mice and the healthy controls, indicating the well tolerability and safety of the combined treatment in mice (Supplementary Fig. 30). Collectively, these results suggested that

IPI549@HMP-augmented RT plus aPD-L1 treatment could further potentiate the generation of a robust synergistic antitumor immune responses, which are essential to elicit abscopal effect.

**Long-term immune memory effects**. As the hallmark of adaptive immunity, immunological memory response can provide long-term protection for organisms against the second pathogen attack, and it is quite critical for tumor prevention. To evaluate the immune memory effects generated by IPI549@HMP-augmented RT plus aPDL1 treatment, a secondary CT26 tumor were inoculated on 46 days after combination therapy, with age- and sex-matched naive mice inoculated with equal number of cancer cells as controls (Fig. 8a). As expected, rapid tumor progression was observed in all naive mice after being challenged with CT26 tumor cells (Fig. 8b–d and Supplementary Fig. 31). In marked contrast, no visible secondary tumor growth was observed in mice after their initial tumors were eliminated by the combined treatment, and all survived for 100 days, demonstrating the remarkable immune memory effect after such proposed combined immunotherapy to enable long-lasting protection of mice from tumor recurrence (Fig. 8e).

To further confirm the underlying mechanism of the above phenomenon, spleens of mice were collected 14 days after tumor rechallenge to determine changes in memory T cells. Antigen-specific memory T cells are classified into central memory (Tcm) and effector memory T cells (Tem) subsets, and Tem subsets (CD3⁺CD8⁺CD62L⁻CD44⁺) located in lymphoid and non-lymphoid tissues can respond quickly when exposed to the same tumor antigen and provide immediate protections by producing cytokines (*e.g.*, TNF-α and IFN-γ). Intriguingly, an obvious shift of Tcm in both CD8⁺ T cells and CD4⁺ T cells towards Tem phenotype was observed in long-term surviving mice after PI549@HMP-augmented RT + aPD-L1 treatment, compared to naive mice (Fig. 8h–j). Likewise, in the combination treatment group, much higher levels of serum cytokines like IFN-γ and TNF-α that play vital roles in cellular immunity against cancer were observed (Fig. 8f, g). Hence, these findings provided critical evidence that a strong antitumor immune response has been established.

## Discussion
Here, relying on a well-established postsurgical model, we offer direct evidence that the local biological consequences of SR can trigger substantial outgrowth of deposited tumor cells at the resection anatomical site. Specifically, SR-induced tumor outgrowth related to a local complex inflammatory reaction characterized by hypoxia and mobilization of myeloid cells as well as the exhaustion of CTL, which ultimately promotes the development of immunosuppressive TME and contributes to the residual tumor escape. Hence, we rationally designed and developed a distinct IPI549@HMP nanomedicine with efficient tumor-homing capacity, TME-responsive drug release ability and in situ oxygen production properties that can enhance local X-ray irradiation-initiated cancer cell killing. Specifically, this IPI549@HMP-based RT could effectively reshape the SR-induced immunosuppressive TME via myeloid cell-targeted immunomodulatory drugs (IPI549) and RT-mediated immunostimulatory effect to favor antitumor immunities, which can also turn PD-L1 blockade non-responding tumors into responding ones. Such comprehensive treatment strategy (IPI549@HMP + RT plus aPDL1) causes systemic and memory antitumor immunity, which not only efficiently regresses or eradicates both the treated primary tumors and untreated distant tumors, but also completely inhibits tumor rechalenge with negligible systemic toxicity.

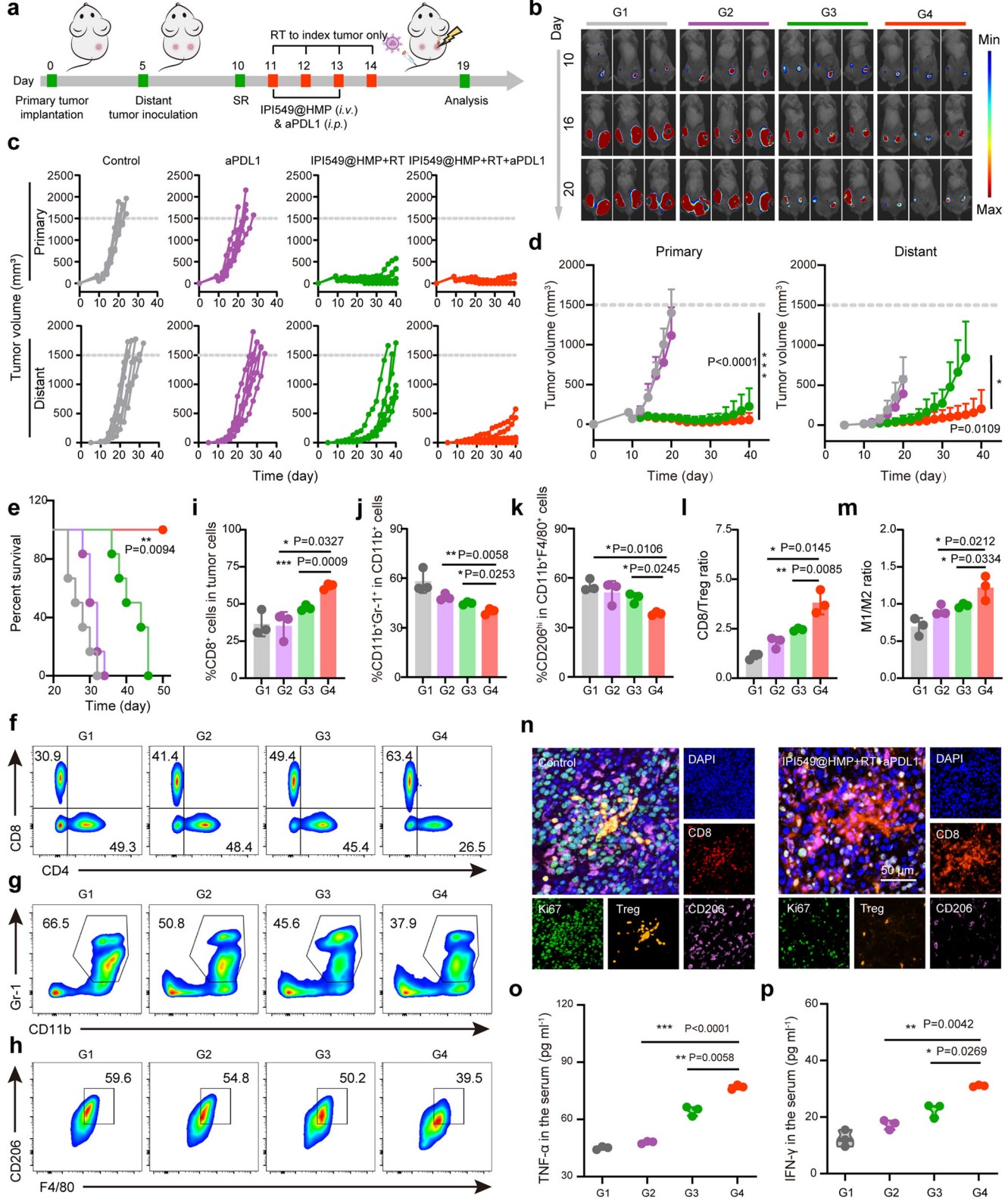

SR-induced wound-healing response contributes to tumor progression, however, the specific mechanisms underlying the rapid outgrowth of residual tumors after SR remain largely obscure. Thus, an advantage of the current study is that we uncover the immune phenotypic changes after SR by using RNAseq and multi-parametric FCM. Our proposed postsurgical setting should likely represent a high-leverage context in which to administer postoperative anticancer immunotherapy. Our study suggests that SR elicits an immunosuppressive and aggressive pattern of residual tumors, mainly owing to enhanced infiltration of myeloid cells in TME. One possibility is that SR generates a sequence of events meant to induce wound healing characterized by the secretion of cytokines, matrix-remodeling enzymes and growth factors[35]. While useful for wound healing, in the context

**Fig. 7 Abscopal effect of IPI549@HMP-augmented RT plus PD-L1 blockade. a** Schematic illustration of the experiment design for in vivo evaluations. **b** Representative bioluminescence images of Luc[+] CT26 tumor of six biologically independent animals from each group after varied treatments as indicated. **c, d** Individual (**c**) and average tumor growth curves (**d**) of primary and distant tumors ($n = 6$ mice in **c, d**). **e** Kaplan-Meier survival curves of mice treated with varied therapeutic combinations ($n = 6$ mice). **f-k** Representative flow cytometric analysis and relative quantification of CTLs (CD8[+]CD3[+]CD45[+]) (**f, i**), MDSCs (CD11b[+]Gr-1[+]CD45[+]) (**g, j**) and TAM-M2 (CD206[hi]CD11b[+]F4/80[+]CD45[+]) (**h, k**). **l, m** Quantification by flow cytometry of CD8/Treg (**l**) and M1/M2 ratios (**m**). **n** Representative polychromatic immunofluorescent staining images of tumor sections from three biologically independent samples showing DAPI (blue), CD8[+] (red), CD206[+] (purple), Foxp3[+] (orange) and Ki67[+] (green) cells infiltration for Control and IPI549@HMP + RT + aPDL1 groups. **o, p** Cytokine levels of TNF-α (**o**) and IFN-γ (**p**) in the serum after varied treatments. G1, Control; G2, aPDL1; G3, IPI549@HMP + RT; G4, IPI549@HMP + RT + aPDL1. RT, radiotherapy; CTLs, cytotoxic T lymphocytes; Tregs, regulatory T cells; MDSCs, myeloid-derived suppressor cells; TAMs-M2, M2-like macrophages; TAMs-M1, M1-like macrophages. Data were expressed as means ± SD ($n = 3$ biologically independent samples in **i–m**, **o** and **p**). Statistical difference was calculated using two-tailed unpaired student's t-test (**d**, **i–m**, **o** and **p**) and Log-rank (Mantel-Cox) test (**e**). *$P < 0.05$, **$P < 0.01$, ***$P < 0.001$. Source data are provided as a Source Data file.

of postsurgical tumors, these mediators contribute to the rapid expansion of MDSCs. Since wound healing after SR is a physiological process of tumor tissue repair, our findings in colorectal cancer may be applicable to other SR-treated solid tumors, such as the accelerated tumor progression observed clinically in patients with gastric carcinoma after invasive diagnostic procedure[36] and the explosive growth of micro-metastatic foci which are not completely removed by SR during the liver regeneration after metastasectomy for colorectal cancer[37]. Using this approach to investigate the specific role of SR in facilitating early tumor progression, we are able to understand and expand upon the small body of literature previously devoted to similar questions.

Given these conclusions, we do not wish to recommend avoiding surgical removal of the tumor because of those potentially negative side effects caused by SR as presented by previous clinical data and proved here experimentally. Instead, bearing on both biological and clinical perspectives, we argue that treatment in the postoperative setting where the primary tumor was incompletely removed is distinct from addressing an intact solid tumor. The postoperative context may represent a consequential timeframe for further intervention, as the concentrated immunosuppression derived from SR must be timely overcome to prevent local tumor recurrence and to improve the efficacy of existing therapies. Specifically, our findings underscore that antagonizing the anticancer immune system in the postoperative setting is sufficient to yield durable survival benefits and induce antitumor memory. Therefore, converting the postoperative period from a pronounced augmenter of tumorigenic effect to a window of opportunity for inhibiting and/or eradicating the residual disease is desired to improve the long-term survival rates.

The fact that myeloid cells are a pivotal resistance mechanism to RT and ICB further emphasizes the urgency of developing therapeutic strategies to effectively target and eliminate these myeloid cells. Previous studies have demonstrated that surgery-generated hypoxia is highly correlated with RT resistance and PI3Kγ is thought to control molecular switch for myeloid cell-triggered immune suppression[37]. In this context, the use of $MnO_2$ nanostructures that alleviate tumor hypoxia and PI3Kγ inhibitor that targeting myeloid cells trafficking can boost RT, thereby conferring sustained tumor regression in postoperative murine models. Particularly, coupling HMP-augmented RT with double blockade of PD-L1 and PI3Kγ can substantially mitigate the consequences of SR and unleash the full potential of antitumor immune responses. The PI3Kγ inhibitor IPI549 is currently undergoing a Phase 1b clinical trial for advanced solid tumors in combination with ICB therapy (clinicaltrials.gov ID NCT02637531). Our results offer strong support for prospective studies testing the impact of postoperative PI3Kγ inhibition on residual tumors and should have profound implications for future postsurgical treatment administration. Considering that complex inflammatory responses

postsurgical may also occur after tumor ablative therapies like microwave ablation (MWA), further application of IPI549@HMP-based RT to solid tumors post-MWA also achieved excellent tumor outgrowth control (Supplementary Fig. 32), demonstrating that this combined radioimmunotherapy strategy has the potential to be extended to post-ablation therapy.

In summary, these findings demonstrate that high infiltration of MDSCs promotes postsurgical immunosuppression TME and provides a potential explanation for SR-induced tumor proliferation, which is the most frequently observed clinic scenario. Accordingly, we engineer a viable therapeutic nanomedicine linking pH-responsive HMP nanoshells with a packed immunomodulator (IPI549) for radiosensitization. It exerts antitumor immunity based on enhanced RT-induced tumor killing and repaired immunosuppression, and also effectively restores the sensitivity of myeloid cell-rich tumors to ICB. Further combination with PD-L1 blockade offers an abscopal response that inhibit primary tumors as well as distant metastasis likely through the CTL migration, and a robust immune-memory effect that effectively protect up to 100% of the treated mice from tumor reimplantation and significantly prolong their survival. Our data provide a strong rationale to consider exploring PI3Kγ inhibitor as adjuvants in battling cancer during perioperative immunosuppression based on a precision-medicine-type assessment of the residual tumor immune landscape.

## Methods
**Materials**. All chemicals in our study were obtained from Sigma-Aldrich unless otherwise specified. Treaethyl orthosilicate (TEOS), poly (allylamine hydrochloride) (PAH, MW: 15, 000), polyacrylic acid (PAA, MW: 2000) and 1-(3-dimethylaminopropyl)−3-ethylcarbodiimide hydrochloride (EDC) were purchased from Aladdin Co., Ltd (Shanghai, China). Amino group terminated glycol (PEG-$NH_2$, MW: 5000) was supplied by Ruixi Biotechnology Co., Ltd (Xi'an, China). IPI549 was obtained from MedChemExpress (Catalog No. HY-100716). In vivo anti-mouse PDL1 antibody was obtained from BioXell (B7-H1, Catalog No. BE0101).

**Cell lines and animals**. The CT26 cell line and B16F10 cell line were originally obtained from American Type Culture Collection (ATCC). All cells were cultured in RMPI 1640 medium (Solarbio) containing 1% penicillin (HyClone), 1% streptomycin (HyClone) and 10% fetal bovine serum (FBS, HyClone) at 37°C in 5% $CO_2$ humidified air. The cells were detected every two months to exclude mycoplasma. Female BALB/c mice and C57BL/6 mice (6-8 weeks) were ordered from Shanghai Salccas Biotechnology Co., Ltd. Mice were housed in an SPF-grade pathogen-free facility with a 12 h light/dark cycle at 20 ± 3°C and a relative humidity of 40% to 70%. All animal experiments were performed according to protocols in accordance with policies of the National Ministry of Health and approved by the Laboratory Animal Center of Shanghai Tenth People's Hospital.

**Postsurgical mouse model construction and RNA sequencing**. A density of $1 \times 10^6$ CT26 cells were subcutaneously injected into female BALB/c mice (6–8 weeks) on the right flank. Ten days after inoculation, the tumor-bearing mice were randomly divided into untreated and SR groups. For SR group, part of the tumor tissue was resected with sterile instruments, leaving about ~80 mm³ of residual tumor tissue. Three days after surgery, total RNA was extracted from the

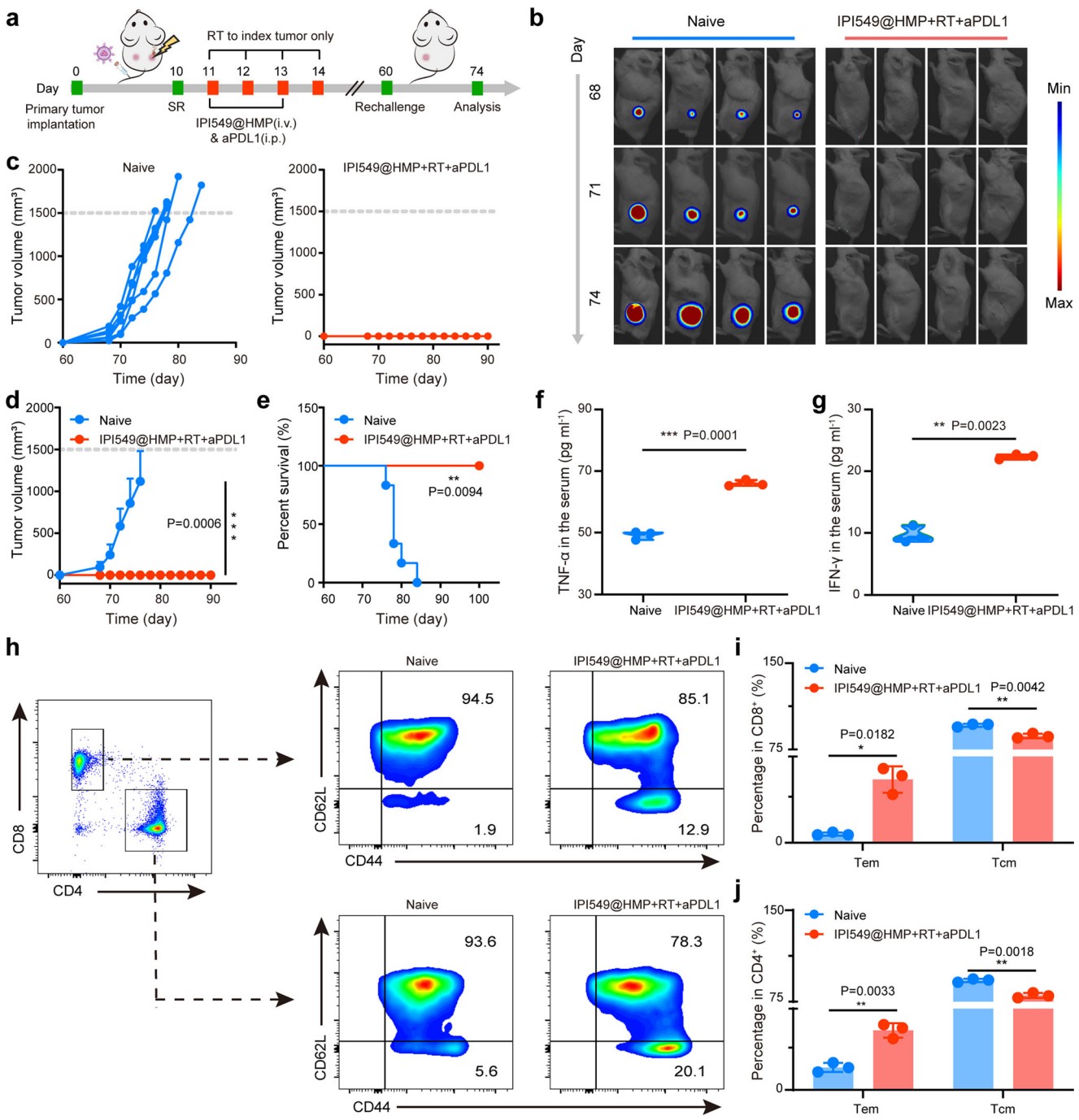

**Fig. 8 Long-term immune memory effects. a** Schematic illustration of the experiment design to assess the immunological memory response triggered by IPI549@HMP-augmented RT and anti-PD-L1 combination therapy. **b** Representative bioluminescence images of Luc$^+$ CT26 tumor of six biologically independent animals from each group after varied treatments as indicated. **c–e** Individual (**c**), average (**d**) tumor growth curves and survival curves (**e**) of the treated mice. Error bars are based on means ± SD ($n = 6$ mice in **c–e**). **f, g** Cytokine levels of TNF-α (**f**) and IFN-γ (**g**) in the serum after tumor rechallenging. **h-j** Representative flow cytometric analysis (**h**) and relative quantification of central memory T cell (Tcm, CD62L$^+$CD44$^+$) and effector memory T cell (Tem, CD62L$^-$CD44$^+$) subset from CD8$^+$ (**i**) and CD4$^+$ (**j**) T cells in the spleen. Data were expressed as means ± SD ($n = 3$ biologically independent samples in **f, g, i** and **j**). Statistical difference was calculated using two-tailed unpaired student's t-test (**d, f, g, i** and **j**) and Log-rank (Mantel-Cox) test (**e**). *$P < 0.05$, **$P < 0.01$, ***$P < 0.001$. Source data are provided as a Source Data file.

tumor tissue of SR-treated tumors and time-matched untreated mice for sequencing. RNA degradation and contamination was monitored on 1% agarose gels. The purity, concentration and integrity of RNA were evaluated with NanoPhotometer® spectrophotometer (IMPLEN, CA, USA), Qubit® RNA Assay Kit in Qubit®2.0 Flurometer (Life Technologies, CA, USA) and RNA Nano 6000 Assay Kit of the Agilent Bioanalyzer 2100 system (Agilent Technologies, CA, USA), respectively. Then, sequencing libraries were generated using NEBNext®UltraTM RNA Library Prep Kit for Illumina®(NEB, USA) following manufacturer's instructions and

index codes were added to attribute sequences to each sample. The quality of the constructed libraries were assessed by Agilent Bioanalyzer 2100 system. The clustering of the index-coded samples was performed on a cBot Cluster Generation System using TruSeq PE Cluster Kit v4-cBot-HS (Illumia) according to the manufacturer's instructions. After cluster generation, the library preparations were sequenced on an Illumina Hiseq platform and paired-end reads were generated for further bio-information analysis. Of note, the P value adjusted by multiple hypothesis test (Padj) was analyzed to control the false-positive ratio.

**Synthesis of IPI549@HMP**. Briefly, TEOS (500 μL) was slowly added to the mixture of ethanol (14 mL), deionized water (2 mL) and ammonia (500 μL) under 45°C water bath stirring for 2 h to obtain sSiO$_2$. After centrifugation with ethanol and water three times, the obtained sSiO$_2$ aqueous solution (15 mg mL$^{-1}$) were mixed with the KMnO$_4$ solution (30 mg mL$^{-1}$) under ultrasound and stirred overnight. The precipitates collected by centrifugation were then dissolved in Na$_2$CO$_3$ solution (2 M) and reacted for 12 h at 60°C to obtain hollow MnO$_2$ (HMnO$_2$). To improve the biological stability, the HMnO$_2$ (2 mg mL$^{-1}$) was modified with PAH (5 mg mL$^{-1}$) and PAA solutions (5 mg mL$^{-1}$) in sequence for 6 h. The above mixture was centrifuged and washed before PEG-NH$_2$ (50 mg) and EDC (20 mg) were added. After stirring for 12 h, the PEG-HMnO$_2$ (HMP) was harvested by centrifugation. For IPI549 loading, the HMP solution and IPI549 were mixed in proportion under magnetic stirring for overnight.

**Characterizations**. Transmission electron microscopy (TEM, FEI Tecnai G2 F30) was used to observe the morphology and elemental composition of nanoparticles. Particle size and zeta potential were measured by a laser granularity sizer (Zetasizer Nano ZS90). UV-vis spectrum was recorded by UV-vis-NIR spectrophotometer (PE Lambda 950). The specific surface area and porosimetry of nanoparticles were analyzed by Accelerated Surface Area and Porosimetry System (ASAP 2460). X-ray photoelectron spectroscopy (XPS, Thermo escalab 250Xi) was adopted to analyze the valence state of manganese ion. Nikon bio-microscope (CI-L) was used to observe the fluorescent sections. Inductively coupled plasma mass spectrometry (ICP-MS, Agilent ICPMS7800) was performed to determine the manganese ion concetration.

**Degradation and drug release studies**. The degradation of HMP nanoparticles in vitro was conducted by mixing HMP with PBS (pH = 7.4), PBS (pH = 6.5) or PBS (pH = 6.5) containing H$_2$O$_2$ (50 or 100 μM) for varied durations at 37°C. At selected time point, the mixture was measured by a UV-vis spectrometer to calculate the degradation rate. For detection of drug release behavior, IPI549@HMP solution was dialyzed with above media at 37°C under shaking. The released drug at varied time points was calculated according to the absorbance of IPI549 at 249 nm.

**In vitro biocompatibility evaluation of HMP**. Cell proliferation toxicity assay was conducted according to the CCK-8 kit (Donjindo Chemical Technology Co., Ltd) instructions. CT26 or B16F10 cells were pre-seeded in 96-well plate overnight. IPI549@HMP solutions at varied concentrations was then added and co-incubated with the cell for 24 h. Subsequently, CCK-8 reagent was added into each well, followed by incubation of 1 h at 37°C. The cell viability was determined by a microplate reader (Thermo Multiskan FC) at the absorbance (OD) of 450 nm. To determine the biocompatibility of IPI549@HMP, hemolysis experiment was performed. First, the fresh red blood cells extracted from healthy female BALB/c mice were prepared into suspension with PBS. Then, the diluted red blood cell suspension was mixed with varied concentrations (6.25, 13.5, 25, 50, 100, 200 ppm) of IPI549@HMP solutions and incubated at 37°C for 2 h. The absorbance of the supernatant at 570 nm was measured and hemolysis (%) was calculated according to the formula: Hemolysis (%) = (OD$_{sample}$ - OD$_{negative}$) / (OD$_{positive}$ - OD$_{negative}$) × 100%. Notably, PBS was set as negative control while deionized water as a positive control.

**Magnetic resonance (MR) imaging**. A 3.0 T MR clinical scanner (United Imaging, uMR 770) with a mouse coil was adopted to monitor imaging performance of IPI549@HMP. The parameters were set as follows: Freq. FOV = 8.0 mm$^2$, phase FOV: 1.00 mm$^2$, Slice thickness: 2.0 mm, Slices: 8, spacing = 0.3 mm, TR: 500 ms, TE: 21 ms. The T$_1$-mapping and T$_1$W$_1$ imaging of IPI549@HMP at varied Mn concentrations (0.64, 0.32, 0.16, 0.08, 0.04 mM) in PBS (pH = 7.4), PBS (pH = 6.5) and PBS (pH = 6.5) containing H$_2$O$_2$ (100 μM) were measured by MR system (uExeed, R002). For tumor imaging, MR scans were conducted before and after intravenous injection of IPI549@HMP (MnO$_2$ = 7.5 mg kg$^{-1}$, IPI549 = 1.5 mg kg$^{-1}$).

**In vivo distribution and pharmacokinetics assays**. CT26 tumor-bearing mice (n = 3) were i.v. injection of ICG-labeled IPI549@HMP when the tumor volume reached about 200 mm$^3$. The fluorescence images were taken at pre-set time points using an in vivo imaging system (IVIS) spectrum imaging system (VISQUE Invivo Smart-LF). The fluorescence intensity (average radiance, photos s$^{-1}$ cm$^{-2}$ sr$^{-1}$) was quantitatively analyzed by living image software. When assessing pharmacokinetic status, 15 μL blood sample was collected from mice at varied time intervals (0.08, 0.16, 0.5, 1, 2, 4, 8, 12, 24 h) post intravenous injection of IPI549@HMP nanoparticles. The half-life time of IPI549@HMP was calculated based on the Mn content determined by ICP-MS in blood samples.

**Assessment of oxygen condition**. The in vitro oxygen production capacity of IPI549@HMP nanoparticles was detected qualitatively and quantitatively by a VEVO ultrasonic imaging and a system portable dissolved oxygen analyzer (Leici JPB-607A). To evaluate oxygen generation capacity in vivo, IPI549@HMP nanoparticles (MnO$_2$ = 7.5 mg kg$^{-1}$, IPI549 = 1.5 mg kg$^{-1}$) were injected into

tumor-bearing mice via the tail vein. The enrichment of nanoparticles and the change of blood oxygen saturation in tumor area were monitored by laser mode and blood oxygen mode of PA imaging system (VEVO LAZR-X) at 0, 2, 4, 8 h post-injection. Blood oxygen saturation is the ratio of PA signal intensity of oxy-hemoglobin (λ = 850 nm) and deoxyhemoglobin (λ = 750 nm). Then, the PA signals of regions of interest (ROI) were measured and analyzed. Tumor tissues were collected at the end of PA signal acquisition and stained with HIF-1α antibody. Notably, five sections were randomly chosen from each group and positive hypoxia area was calculated by Image J software (Version 1.8.0).

**γ-H2AX immunofluorescence assay**. CT26 cells were pre-seeded in six-well plates overnight and incubated with blank/HMP/ IPI549@HMP nanoparticles (100 ppm) for 8 h. X-ray irradiation was then performed at 225 KV and 8 mA for total 6 Gy. After 4% formaldehyde fixing, methanol breaking and 1% BSA blocking, cells were incubated with γ-H2AX antibody (CST, D7T2V, Catalog No.80312 S) diluted at 1:200 at 4°C overnight, followed by incubating with secondary antibody for another 1 h. The cell nuclei were stained with DAPI ten minutes before imaging.

**In vitro cytotoxicity assay**. The in vitro antitumor efficacy of IPI549@HMP-sensitized radiotherapy was evaluated using Calcein-AM/PI Double Staining Kit (Dojindo). Briefly, CT26 cells were cultured in Nunclon Sphera Microplates (Thermo, Catalog No.174925) to obtain multicellular spheroids (MCSs). The formed MCSs were co-incubated with PBS/HMP/IPI549@HMP for 12 h followed by X-ray radiation (225 KV and 8 mA for total 6 Gy) based on groups. After further incubation for 24 h, the MCSs were stained and observed under confocal microscope to estimate cytotoxicity.

**Colony formation assay**. CT26 cells were pre-incubated with PBS/HMP/ IPI549@HMP and irradiated as indicated. One hour after the X-ray irradiation (6 Gy), each group cells were transferred into six-well plates (200 cells per well) and further cultured for 10 days. Once the cell colonies were formed, cells were fixed with 4% paraformaldehyde and stained with Giemsa dye (Leagene Biotechnology). The surviving colonies were identified as colonies containing more than 50 cells.

**In vivo toxicological evaluation**. The female BALB/c mice (6-8 weeks) were intravenously injected with IPI549@HMP nanoparticles (MnO$_2$ = 7.5 mg kg$^{-1}$, IPI549 = 1.5 mg kg$^{-1}$) or 0.9% physiological saline every two days for a total of two times. During the observation, the mice were weighed every other day. On the 10th and 20th day, mice serum and whole blood were extracted for blood biochemical and blood cell analysis, while the main organs were isolated for H&E staining.

**Tumor models and treatment experiments**. To establish unilateral residual tumor model, a density of 1×10$^6$ CT26 or Luc$^+$ CT26 cells were subcutaneously injected into the female BALB/c mice (6-8 weeks) on the right flank. Treatments were initiated when the tumor size reached 150-200 mm$^3$ (10 days after inoculation). To construct the subcutaneous melanoma model, B16F10 cells (1×10$^6$) were injected into the back of each female C57BL/6 mouse (6-8 weeks). On day 7 after inoculation, B16F10 tumor-bearing mice were treated with surgical resection. Before treatment, mice were anesthetized with 1% pentobarbital sodium (100 μL per 15 g body weight). For SR model, most of the tumor tissue was resected with sterile instruments, leaving about 50-80 mm$^3$ of residual tumor tissue and the wound was then sutured. Mice were then randomly divided into five groups, including control group, IPI549@HMP group, RT group, HMP + RT group, IPI549@HMP + RT group. Eight hours before RT, mice in different groups were injected with PBS/HMP/ IPI549@HMP (MnO$_2$ = 7.5 mg kg$^{-1}$, IPI549 = 1.5 mg kg$^{-1}$) via the tail vein. For groups containing RT, mice were placed in separate lead box so that the tumor area was exposed to the cut-off for irradiation, while the rest of the body was completely shielded. X-ray irradiation was performed on X-ray irradiator (Rad Source RS2000), of which the parameters were set at 225 KV and 8 mA. The optimized radiation dose was conducted with two successive 3 Gy irradiation at 8 h post-injection for two cycles. The tumor size and body weight of mice were measured every other day after treatment. The tumor volume (mm$^3$) was calculated according to (length × width$^2$) × 0.5. In vivo fluorescence imaging was used to track tumor burden in real time. D-Luciferin (15 mg mL$^{-1}$, 150 μL per 20 g body weight) was intraperitoneally injected into each mouse eight minutes before anesthetized with isoflurane (3% for induction and 1.5% for maintenance). When the tumor volume exceeded 1500 mm$^3$ or cachexia signs appeared, mice were euthanized.

To establish microwave ablation tumor model, microwave ablation was performed using a cooled-tip electrode with 0.5 cm needle tip inserted percutaneously. Ablation lasted for 1~2 min at 5 W resulted in partial necrosis of the tumor tissue. Mice were then randomly divided into two groups, including control group and IPI549@HMP + RT group. Eight hours before RT, mice in different groups were injected with PBS or IPI549@HMP (MnO$_2$ = 7.5 mg kg$^{-1}$, IPI549 = 1.5 mg kg$^{-1}$) via the tail vein. X-ray irradiation parameters were set at 225 KV and 8 mA and the optimized radiation dose was conducted with two successive 3 Gy irradiation at 8 h post-injection for two cycles.

To establish bilateral tumor model, a distant tumor (1×10$^6$ CT26 or Luc$^+$ CT26 cells) was transplanted into the left flank of female BALB/c mice (6–8 weeks) 5 days

after the right flank tumor was inoculated. Ten days after primary tumor establishment, most of the tumors in the right flank were resected leaving about 50-80 mm³ of residual tumor tissue. Then the mice were randomly divided into four groups including control group, aPDL1 group, IPI549@HMP + RT group and IPI549@HMP + RT + aPDL1 group. PBS or IPI549@HMP ($MnO_2$ = 7.5 mg kg$^{-1}$, IPI549 = 1.5 mg kg$^{-1}$) were intravenously injected on 1$^{st}$ and 3$^{rd}$ day postoperation. And anti-PD-L1 (3.75 mg kg$^{-1}$) was intraperitoneally administered to mice every 2 days for a total of twice. The subsequent monitoring of bilateral tumors and survival duration was the same as aforementioned procedures.

**T cell depletion experiments**. Female BABL/c mice bearing CT26 tumors were treated with IPI549@HMP + RT after surgery, as described before. The treated mice were *i.p.* injected with in vivo anti-mouse CD8 (10 mg kg$^{-1}$, BioXell, 53-6.7, Catalog No.BE0004) or in vivo anti-mouse CD4 (10 mg kg$^{-1}$, BioXell, GK1.5, Catalog No.BE0003) on day 10, 13, and 16 after tumor inoculation. Peripheral T cell depletion was confirmed by flow cytometry.

**Western blot procedures**. The equivalent protein (quantitative with a bicinchoninic acid protein assay kit, Beyotime, Catalog No.P0012) was mixed with the protein loading buffer (Tanon$^{TM}$, Catalog No.180-8201 A) and boiled at 100℃ for 10 min. After gel electrophoresis and protein transfer, the protein was incubated with 1:1000 diluted anti-PI3K p110 gamma antibody (CST, D55D5, Catalog No.5405 T) and anti-GADPH antibody (CST, D16H11, Catalog No.5174 T) at 4℃ overnight followed by incubation with 1:5000 diluted anti-rabbit IgG-HRP secondary antibody (CST, L27A9, Catalog No.5127 S) for 1 h at room temperature. Electrochemiluminescence imaging was captured by an automatic chemiluminescence image analysis system (Tanon 4600).

**FCM analysis**. The isolated tumor tissues or spleens were digested into single-cell suspension and pre-incubated (15 min, 4 ℃) with anti-CD16/32 (eBioscience, FRC-4G8, Catalog No. MFCR00) monoclonal antibody to block nonspecific binding and then stained with diluted fluorochrome-conjugated antibodies. The antibodies involved in the experiment include CD45-eF506 (eBioscience, 30-F11, Catalog No.69-0451-82), CD3-PE-cy7 (eBioscience, 145-2C11, Catalog No.25-0031-82), CD4-FITC (eBioscience, GK1.5, Catalog No.11-0041-82), CD8-Percp-cy5.5 (eBioscience, 53-6.7, Catalog No.45-0081-82), Foxp3-PE (eBioscience, FJK-16s, Catalog No.12-5773-82), CD11b-PE-cy7 (eBioscience, M1/70, Catalog No.25-0112-82), CD11c-PE (eBioscience, N418, Catalog No.12-0114-82), MHC-II-eF450 (eBioscience, M5/114.15.2, Catalog No.48-5321-82), Ly6c-percp-Cy5.5 (eBioscience, HK1.4, Catalog No.45-5932-82), Ly6G-PE (eBioscience, 1A8-Ly6g, Catalog No.12-9668-82), Gr-1-APC (Biolegend, RB6-8C5, Catalog No.108412), F4/80-BV421 (Biolegend, BM8, Catalog No.123137), CD206-FITC (Biolegend, C068C2, Catalog No.141703), CD80-APC (Biolegend, 16-10A1, Catalog No.104714), CD62L-APC (eBioscience, MEL-14, Catalog No. 17-0621), CD44-PE (eBioscience, IM7, Catalog No. 12-0441). Other reagents, including the RBC lysis buffer (Catalog No.00-4300-54), intracellular fix/perm buffer set (Catalog No.88-8824-00) and Foxp3 transcription factor staining buffer set (Catalog No.00-5523-00) were obtained from Thermo. All antibodies were diluted to the working concentration of 0.2 µg per test. Finally, the stained cells were filtered and detected by FACS FCM (BD, Fortessa X20) and analyzed by Flowjo software (TreeStar, 10.6.2).

**Polychromatic immunofluorescent staining**. The as-prepared tumor sections were stained according to the instructions of five-color multiplex fluorescence immunohistochemical staining kit (Absin, Catalog No.abs50013) and blocked with TBST containing 5% goat serum before incubation with antibodies. The antibodies involved in experiment include CD45 (abcam, Catalog No.ab10558, diluted at 1:500), CD11b (abcam, EPR1344, Catalog No.ab133357, diluted at 1:4000), Ki67 (abcam, Catalog No.ab15580, diluted at 1:500), HIF-1α (Bioss, Catalog No.bs-0737R, diluted at 1:200), CD206 (CST, E6T5J, Catalog No.24595, diluted at 1:200), CD8 (CST, D4W2Z, Catalog No.98941, diluted at 1:200), Foxp3 (abcam, EPR22102-37, Catalog No.ab215206, diluted at 1:200), HMGB1 (CST, Catalog No.3935, diluted at 1:100), calreticulin (CST, D3E6, Catalog No.12238, diluted at 1:400) and anti-NKp46 (Abcam, EPR23097-35, Catalog No. ab233558, diluted at 1:500). The nuclei were stained with DAPI before sealing, and all sections were scanned by a fluorescent scanning camera (KFBIO, KF-TB-400).

**Statistical analyses**. The quantitative data were expressed as mean ± SD. Statistical differences were calculated by GraphPad Prism software (version 8) using unpaired student's *t*-test or one-way analysis of variance (ANOVA) when comparing two or multiple groups, respectively. The survival curve was analyzed by the Log-rank test. Statistical differences are denoted as ns, not significant. *$P < 0.05$, **$P < 0.01$, ***$P < 0.001$.

**Reporting summary**. Further information on research design is available in the Nature Research Reporting Summary linked to this article.

## Data availability
The raw sequencing data generated in this study have been deposited in the Genome Sequence Archive (GSA) database under accession code CRA006269. The remaining data are available within the Article, Supplementary Information or Source Data file.

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

## Acknowledgements

This work was supported in part by the National Natural Science Foundation of China grant to H.X. (81725008 and 81927801), and W.Y. (82171943), Science and Technology Commission of Shanghai Municipality grant to H.X. (21Y21901200 and 19DZ2251100), Shanghai Municipal Health Commission grant to H.X. (2019LJ21 and SHSLCZDZK03502), Scientific Research and Development Fund of Zhongshan Hospital of Fudan University grant to H.X. (2022ZSQD07) and Shanghai Rising-Star Program grant to W.Y. (21QA1407200).

## Author contributions

W.Y., H.X. and Y.C. designed and supervised the project and commented on the project. X.G., L.S., Y.S., W.Y., F.J., X.L., X.B., X.H. and C.Z. performed the experiments. X.G analyzed the data. W.Y. and X.G. wrote the manuscript. All the authors contributed to the discussion during the whole project.

## Competing interests

The authors declare no competing interests.
