## [Peer Review File · Nature Communications]

Reviewers' Comments:

Reviewer #1:

Remarks to the Author:

In this manuscript, Guan et al presented the genomic landscape of tumor after surgical resection and demonstrated that the myeloid cell-mediated immunosuppressive effects play a vital role in fostering tumor recurrence and hindering PD-L1 blockade post-surgery. Intriguingly, the focus of this work is a clinical problem that needs to be solved urgently. The authors described a highly efficient combined tumour-therapeutic modality based on the rational integration of nanomedicine-augmented radiotherapy and checkpoint-blockade immunotherapy. IPI549 was capable of targeting myeloid cells and disrupting the immunosuppressive niche, subsequently improving ICB-mediated antitumor immune response. This radioimmunotherapy nanosystem (IPI549@HMP) in combination with PD-L1 blockade could mimic a "hot" tumor-immunity niche to inhibit tumor progression, metastasis and rechallenge. Overall, it is a piece of excellent work. I recommend it is accepted after addressing the following minor issues:

- 1) The degradation of HMP was evaluated in the presence of PBS with 100 μ M H₂O₂ at 37°C. However, the concentration of H₂O₂ (100 μ M) was a little higher. Thus, a more moderate H₂O₂ level should be applied to mimic the conditions in tumor tissues in vitro.
- 2) Gene ontology analysis could be useful data that can explain various immune related events. More details should be stated for the clarification.
- 3) While the findings are interesting and extensive on CT26 tumor model, other tumor cell lines may be used to evaluate the generality of this combined radioimmunotherapy strategy.
- 4) In Figure 5b, Figure 7b, Figure 8b, the authors utilized fluorescence imaging, and I would like to suggest authors to calculate the light intensities and do the statistics, which can more objectively reflect the treatment differences.
- 5) The complex inflammatory reaction after surgery can also occur post-tumor ablation therapies, such as microwave ablation, radiofrequency ablation and high-intensity focused ultrasound (HIFU) therapy. It is not clear if this nanosystem apply to solid tumors treated with ablation procedures.
- 6) The biocompatibility and biosafety of the adopted bio-responsive nanosystem should be clarified in detail for guaranteeing further potential clinical translation.

Reviewer #2:

Remarks to the Author:

In this manuscript, the authors established a tumor model of cancer recurrence after surgical resection (SR) and found the recurrence tumor has a more suppressive and hypoxic microenvironment with increased myeloid cells infiltration. To address the outgrowth of recurrent tumor in this model, the authors developed a PEG-HMnO₂- (HMP)-bridged nanoparticle loading with PI3ky inhibitor (IPI549), the engineered nanoparticle (IPI549@HMP) could target myeloid cells, and catalyzing endogenous H₂O₂ into O₂ to achieve hypoxia-relieved-radiotherapy (RT). Intravenous injection of the particle plus radiotherapy can induce tumor regression through reprogramming the tumor microenvironment and can synergize with anti-PD-L1 therapy. Moreover, the long-term memory and abscopal effect were observed when the IPI549@HMP plus RT combined with anti-PD-L1 blockade. Overall, the data demonstrate the therapeutic potential of the engineered nanoparticle in post-surgical cancer recurrence and provide a new direction to manipulate the tumor microenvironment for cancer therapy.

Concerns:

1. How about the pharmacokinetics and biodistribution of the IPI549@HMP? Does it also accumulate in organs, especially liver more than tumors and any toxicity after systemic delivery?
2. In this study, all the in vivo data showed in the manuscript were based on luciferase+ CT26 tumor, but the luciferase has strong immunogenicity and the tumor regression could be driven by luciferase-specific T cells. Does the combination therapy (IPI549@HMP plus RT) also effective in wild type CT26 and other tumor types?
3. The efficacy of IPI549@HMP treatment depends on DC or MDSC?
4. The percentage of TAMs and MDSCs have decreased after combination therapy, Are they killed by CTL?
5. PI3K- γ inhibitor can block neutrophil or MDSC recruitment, does any changes observed in the

the population in the combination therapy?

6. Which subsets of T cells were necessary for tumor control? CD8+T cells or CD4+T cells? Do NK cells also essential for tumor control in combination therapy?

Reviewer #3:

Remarks to the Author:

In this manuscript, Gan et al. developed a PI3Kγ-targeted nanosystem, established an experimental colon cancer model for post-surgical immuno-radiotherapy, and demonstrated a robust tumor control and immune memory. The work is original and of significance to its field. The experiments are adequate and well-designed. The methodology is well-structured and supported with comprehensive visuals, leading up to comprehensible and proper conclusions, although some of the analysis could use more expounding and details.

However, I would like to elaborate on one major concern I am left with and offer two suggestions.

My major concern lies in the establishment of the surgical resection (SR) model system, which is the foundation of this study. In Figure 1, the authors compared post-SR tumors mainly with untreated tumors. Fig. 1b showed significant larger volumes of post-SR tumors on day 20 to day 24, as compared to untreated or sham operation tumors. It is unclear, however, how the authors adjust the start point of the growth curve among three treatment groups. On day 10, for example, the post-SR residual tumors are of a smaller size (50-80 mm³) than the untreated tumors (150-200 mm³), which is not reflected in Fig. 1b. On page 7, the authors stated "...compared with size-matched tumors (line 119)". To match the size of the tumor in different treatment groups, the curves must start at different time (dates) due to the surgical removal of a large proportion of the tumor in the SR group, which is, again, not reflected in Fig. 1b. It is critically important to clarify (in "Methods", "Results", and "Figure Legends") the start point of comparison, as size-matching may result in an inappropriate comparison between a late-stage faster-growing post-SR tumors with early-stage slower-growing untreated tumors. As the follow-up experiments in Fig. 1d-1q implemented the same comparison strategy, an inappropriate design or a lack of justification regarding the comparison strategy leads to questioning the credibility of all these transcriptomics and immune profiling data.

My first suggestion is in relation to my stated concern. To establish post-SR tumors as the specific target to benefit from the IPI549@HMP+RT regimen, the authors would better compare post-SR tumors with untreated tumors for their responses to IPI549@HMP+RT. In other words, I suggest that experiments in Figure 5 be repeated with non-SR (untreated) tumors to strengthen the conclusion that post-SR tumors exhibit a better response to IPI549@HMP+RT than non-SR (untreated) tumors. In these new experiments, the author would have to clarify the treatment starting time for both groups, i.e., whether to match the tumor size or the time after tumor inoculation (tumor-growth phase).

My second suggestion is related to the abscopal effect of IPI549@HMP+RT (Fig. 7). Fig. 7c showed no inhibitory effects of anti-PD-L1 alone, whereas IPI549@HMP+RT+aPD-L1 elicited a robust inhibition on either primary or abscopal tumors. These results suggest that IPI549@HMP+RT creates an enhanced vulnerability to anti-PD-L1, likely due to an induction of PD-L1 expression. I recommend a comparison of PD-L1 expression in both primary and abscopal tumors before and after IPI549@HMP+RT treatment.

Response to Reviewer 1

Comments from Reviewer #1 (expertise: MnO₂ based nanoparticles):

In this manuscript, Guan et al presented the genomic landscape of tumor after surgical resection and demonstrated that the myeloid cell-mediated immunosuppressive effects play a vital role in fostering tumor recurrence and hindering PD-L1 blockade post-surgery. Intriguingly, the focus of this work is a clinical problem that needs to be solved urgently. The authors described a highly efficient combined tumour-therapeutic modality based on the rational integration of nanomedicine-augmented radiotherapy and checkpoint-blockade immunotherapy. IPI549 was capable of targeting myeloid cells and disrupting the immunosuppressive niche, subsequently improving ICB-mediated antitumor immune response. This radioimmunotherapy nanosystem (IPI549@HMP) in combination with PD-L1 blockade could mimic a "hot" tumor-immunity niche to inhibit tumor progression, metastasis and rechallenge. Overall, it is a piece of excellent work. I recommend it is accepted after addressing the following minor issues:

Response: Thank you very much for the positive comments and constructive suggestions. Please find the following detailed responses to your comments and suggestions.

(1) The degradation of HMP was evaluated in the presence of PBS with 100 μM H₂O₂ at 37°C. However, the concentration of H₂O₂ (100 μM) was a little higher. Thus, a more moderate H₂O₂ level should be applied to mimic the conditions in tumor tissues *in vitro*.

Response: Thank you very much for the constructive suggestion, which is highly appreciated. Considering the existence of endogenous H₂O₂ with concentrations in the range of 10-100 μM inside most types of solid tumors, we chose 100 μM H₂O₂¹⁻⁵ to mimic the condition within tumor microenvironment. According to the reviewer's suggestion, the degradation of HMP was further evaluated in the presence of PBS with 50 μM H₂O₂ at 37°C. As shown in **Supplementary Fig. 6**, the HMP nanoparticles showed no significant change in neutral environment (pH 7.4) even after 24 h, while they exhibited time-dependent degradation behavior in an acidic environment (pH 6.5) mainly attributing to the decomposition of MnO₂ into Mn²⁺ ions. Remarkably, the release amount of Mn²⁺ increased dramatically after addition of H₂O₂ and the degradation behavior was

positively correlated with the H₂O₂ concentration.

We have clarified this issue in the revised manuscript and highlighted it in yellow. (Lines 192-197 of Page 9, Revised Manuscript and Page 7, Revised Supplementary Information).

Supplementary Fig. 6 Degradation behaviors of HMP. a UV-vis spectrum and corresponding digital photos of HMP aqueous solution after reaction with varied media as indicated. **b** Accumulated degradation profiles of HMP dispersed in PBS (pH 7.4), PBS (pH 6.5) and PBS (pH 6.5) containing H₂O₂ (50 μM or 100 μM). Data were denoted as the mean ± SD (n = 3).

Reference:

1. Yang G, *et al.* Hollow MnO₂ as a tumor-microenvironment-responsive biodegradable nano-platform for combination therapy favoring antitumor immune responses. *Nat Commun* **8**, 902 (2017).
2. Zhu Y, *et al.* Stimuli-Responsive Manganese Single-Atom Nanozyme for Tumor Therapy via Integrated Cascade Reactions. *Angew Chem Int Edit* **60**, 9480-9488 (2021).
3. Huo M, Wang L, Wang Y, Chen Y, Shi J. Nanocatalytic Tumor Therapy by Single-Atom Catalysts. *ACS Nano* **13**, 2643-2653 (2019).
4. Wang C, *et al.* Photosensitizer-Modified MnO₂ Nanoparticles to Enhance Photodynamic Treatment of Abscesses and Boost Immune Protection for Treated Mice. *Small* **16**, 2000589 (2020).
5. Zeng W, *et al.* Dual-response oxygen-generating MnO₂ nanoparticles with polydopamine

modification for combined photothermal-photodynamic therapy. *Chem Eng J* **389**, 124494 (2020).

(2) Gene ontology analysis could be useful data that can explain various immune related events. More details should be stated for the clarification.

Response: Thank you very much for the constructive suggestion, which is highly appreciated. Gene Ontology (GO) analysis revealed that the differentially expressed genes were significantly enriched in the biological process (BP) of immune-related functions and pathways, mainly including immune system process, defense response, immune response and regulation of response to stimulus, suggesting a strong correlation between SR and immune-related functions. Meanwhile, they were enriched in molecular functions (MF) associated with receptor-binding such as protein binding and signaling receptor binding (**Fig. 2f**). We have added more details about gene ontology analysis in the revised manuscript, which is marked in yellow. (**Lines 110-115 of page 5-6, Revised Manuscript**).

(3) While the findings are interesting and extensive on CT26 tumor model, other tumor cell lines may be used to evaluate the generality of this combined radioimmunotherapy strategy.

Response: Thank you very much for the constructive suggestion. According to the reviewer's suggestion, we further explored whether such a combined radioimmunotherapy strategy could potentially be used for other tumor types by employing murine B16F10 melanoma as the model in addition to the colon CT26 cancer model. Similar to previous observations, IPI549@HMP+RT also exhibited remarkable therapeutic efficacy against murine postsurgical residual melanoma and significantly reduced the tumor growth rate compared to controls (**Supplementary Fig. 17**). These results successfully proved that our combined radioimmunotherapy strategy in CT26 cancer model can be extended to other types of tumors. (**Lines 287-293 of Page 14, Revised Manuscript and Page 18, Revised Supplementary Information**).

Supplementary Fig. 17 IPI549@HMP-augmented RT against postsurgical melanoma. a Schematic illustration of the experiment design to assess the *in vivo* IPI549@HMP-based RT in B16F10 melanoma. **b** Residual tumor growth kinetics of mice after varied treatments (n = 6). **c** Weight of the excised tumors examined on day 15 after varied treatments (n = 6). **d** Digital photos of the excised tumors examined on day 15 after varied treatments. SR, surgical resection, RT, radiotherapy. Data were expressed as means \pm SD. Statistical difference was calculated using unpaired student's *t*-test, ***P<0.001.

(4) In Figure 5b, Figure 7b, Figure 8b, the authors utilized fluorescence imaging, and I would like to suggest authors to calculate the light intensities and do the statistics, which can more objectively reflect the treatment differences.

Response: Thank you very much for the constructive suggestion. In order to reflect the treatment differences more objectively, all fluorescence images involved in the manuscript were quantified and statistically analyzed as requested by the reviewer. (Page 16, 27 and 32, Revised Supplementary Information).

(5) The complex inflammatory reaction after surgery can also occur post-tumor ablation therapies, such as microwave ablation, radiofrequency ablation and high-intensity focused

ultrasound (HIFU) therapy. It is not clear if this nanosystem apply to solid tumors treated with ablation procedures.

Response: We gratefully appreciate your valuable suggestion. To elucidate whether this nanosystem is suitable for solid tumors treated with ablation procedures, we constructed a post-microwave ablation (MWA) tumor model. MWA was performed on the 10th day after tumor inoculation causing partial necrosis of the tumor. The residual tumors were then randomized into the Control (PBS) group or IPI549@HMP+RT group (dose of MnO₂ = 7.5 mg kg⁻¹ and IPI549 = 1.5 mg kg⁻¹) treated in the same manner as before (**Supplementary Fig. 32a**). All mice in the control group died within 30 days, while mice in the IPI549@HMP+RT group survived more than 40 days, and even 50% (3/6) of the mice were tumor-free (**Supplementary Fig. 32b-f**). Notably, there was no abnormality in the body weight of combined treatment group compared to control group, except for a transient weight loss during treatment period mainly due to frequent anesthesia, (**Supplementary Fig. 32g**), further demonstrating the efficacy as well as safety of our combined radioimmunotherapy strategy and its suitability for post-ablation tumors. We have added this part in the revised manuscript, which is marked in yellow. (**Lines 463-467 of Page 22, Revised Manuscript and Page 33, Revised Supplementary Information**).

(6) The biocompatibility and biosafety of the adopted bio-responsive nanosystem should be clarified in detail for guaranteeing further potential clinical translation.

Response: Thank you very much for the constructive suggestion, which is highly appreciated. **Firstly**, the two main components within the designed nanosystem (IPI549@HMP) are all validated and the dose of these components used in this study had already been systematically verified in previous studies¹⁻⁴. Notably, the PI3K γ inhibitor IPI549 is currently undergoing a Phase 1b clinical trial for advanced solid tumors in combination with ICB therapy (clinicaltrials.gov ID NCT02637531). Therefore, the intrinsic toxicity of the designed nanoparticles is theoretically low. **Secondly**, ahead of biomedical applications, the potential toxicity of the nanomaterials was evaluated systematically. Cytotoxicity tests were performed, in which cells were incubated with a variety of concentrations of nanoparticles for a certain range of times and then the viability of cells was measured. As presented in **Supplementary Fig. 9**,

IPI549@HMP nanoparticles exhibit high cell viability under 100 μ M concentration after co-incubation with CT26 and B16F10 cells, suggesting they possess low cytotoxicity and good biocompatibility. Meanwhile, the blood cell hemolysis rate was still lower than 5% even at HMP concentration of 200 ppm (**Fig. 3k**), demonstrating that this nanosystem can be safely administered intravenously. **Thirdly**, to further analyze the potential toxicity of IPI549@HMP *in vivo*, a long term safety evaluation and pathological analysis were executed, such as body weight, histological, serum biochemical assays and tissue biodistribution. When mice were *i.v.* injected with ICG-labeled IPI549@HMP, a time-dependent clearance effect mainly by the liver and kidney could be seen (**Supplementary Fig. S11a-d**). Meanwhile, the blood-circulation half-time was calculated to be 0.97 h (**Supplementary Fig. S11e**), which indicates the easy elimination of the IPI549@HMP from the central chamber, such as kidney and liver. Furthermore, serum biochemical assays showed no significant variations and the hematoxylin and eosin (H&E) staining of the major organs (heart, liver, spleen, lung, and kidney) also exhibited no obvious pathological abnormalities compared with the normal mice, indicating the excellent histocompatibility of IPI549@HMP (**Supplementary Fig. 10**). All the *in vitro* and *in vivo* toxicity results suggested that the adopted bio-responsive nanosystem did not show evident toxic effects under a certain concentration, further ensuring its potential clinical translation. (**Lines 221-231 of Page 10-11, Revised Manuscript and Page 12, Revised Supplementary Information**).

Reference:

1. De Henau O, *et al.* Overcoming resistance to checkpoint blockade therapy by targeting PI3Kgamma in myeloid cells. *Nature* **539**, 443-447 (2016).
2. Kaneda MM, *et al.* PI3Kgamma is a molecular switch that controls immune suppression. *Nature* **539**, 437-442 (2016).
3. Yang G, *et al.* Hollow MnO₂ as a tumor-microenvironment-responsive biodegradable nano-platform for combination therapy favoring antitumor immune responses. *Nat Commun* **8**, 902 (2017).

4. Wang C, *et al.* Photosensitizer-Modified MnO₂ Nanoparticles to Enhance Photodynamic Treatment of Abscesses and Boost Immune Protection for Treated Mice. *Small* **16**, 2000589 (2020).

Response to Reviewer 2

Comments from Reviewer #2 (expertise: Radioimmunotherapy):

In this manuscript, the authors established a tumor model of cancer recurrence after surgical resection (SR) and found the recurrence tumor has a more suppressive and hypoxic microenvironment with increased myeloid cells infiltration. To address the outgrowth of recurrent tumor in this model, the authors developed a PEG-HMnO₂-(HMP)-bridged nanoparticle loading with PI3K γ inhibitor (IPI549), the engineered nanoparticle (IPI549@HMP) could target myeloid cells, and catalyzing endogenous H₂O₂ into O₂ to achieve hypoxia-relieved-radiotherapy (RT). Intravenous injection of the particle plus radiotherapy can induce tumor regression through reprogramming the tumor microenvironment and can synergize with anti-PD-L1 therapy. Moreover, the long-term memory and abscopal effect were observed when the IPI549@HMP plus RT combined with anti-PD-L1 blockade. Overall, the data demonstrate the therapeutic potential of the engineered nanoparticle in post-surgical cancer recurrence and provide a new direction to manipulate the tumor microenvironment for cancer therapy.

Response: Thank you very much for the kind comment and suggestion. Please find the following detailed responses.

(1) How about the pharmacokinetics and biodistribution of the IPI549@HMP? Does it also accumulate in organs, especially liver more than tumors and any toxicity after systemic delivery?

Response: Thank you very much for the kind question, which is highly appreciated. We have supplemented the *in vivo* distribution and pharmacokinetic results of IPI549@HMP as suggested by the reviewer in the revised manuscript, and marked in yellow. **(Lines 221-231 of Page 10-11, Revised Manuscript and Page 12, Revised Supplementary Information).**

The distribution and tumor accumulation of IPI549@HMP in CT26 tumor-bearing mice were evaluated by tracking the fluorescence of ICG-labeled IPI549@HMP using an IVIS spectrum imaging system. As shown in **Supplementary Fig. S11a, b**, the ICG fluorescence intensity in the tumor area increased with time and reached at a peak level at 8 h after injection.

Semi-quantitative biodistribution according to isolated major organs indicated high tumor uptake and retention of IPI549@HMP (**Supplementary Fig. S11c, d**). Notably, distinct fluorescence in the liver and kidneys was presented as expected, since the platform starts to degrade and be excreted over this time period. Meanwhile, the blood-circulation half-time was calculated to be 0.97 h (**Supplementary Fig. S11e**), which indicates the easy elimination of the IPI549@HMP from the central chamber, such as kidney and liver.

Ahead of biomedical applications, it is very crucial and necessary to conduct a potential toxicity analysis of nanomaterials. Cytotoxicity tests were performed, in which cells were incubated with a variety of concentrations of nanoparticles for a certain range of times and then the viability of cells was measured. As presented in **Supplementary Fig. S9**, IPI549@HMP exhibit high cell viability under 100 μ M concentration after co-incubation with CT26 and B16F10 cells, suggesting that they possess low cytotoxicity and good biocompatibility. Meanwhile, the blood cell hemolysis rate was still lower than 5% even at HMP concentration of 200 ppm (**Fig. 3k**), demonstrating that this nanosystem can be safely administered intravenously. Moreover, to analyze the potential toxicity after systemic delivery of IPI549@HMP *in vivo*, a long term safety evaluation and pathological analysis were executed, such as body weight, histological, and serum biochemical assays. The serum biochemical assays showed no significant variations and the hematoxylin and eosin (H&E) staining of the major organs (heart, liver, spleen, lung, and kidney) also exhibited no obvious pathological abnormalities compared with the normal mice, indicating the excellent histocompatibility of IPI549@HMP (**Supplementary Fig. S10**). All the *in vitro* and *in vivo* toxicity results suggested that the adopted bio-responsive nanosystem did not show evident toxic effects under a certain concentration, further ensuring potential clinical translation.

Supplementary Fig. 11 *In vivo* biodistribution and pharmacokinetics evaluation of IPI549@HMP. **a, b** *In vivo* fluorescence images and relative quantification of CT26 tumor-bearing mice taken at pre-set time points post *i.v.* injection of IPI549@HMP. **c, d** *Ex vivo* fluorescence images and relative quantification of major organs and tumor dissected from mice at 24 h and 48 h. **e** Blood circulation curve of Mn concentration after *i.v.* administration of IPI549@HMP.

(2) In this study, all the *in vivo* data showed in the manuscript were based on luciferase⁺ CT26 tumor, but the luciferase has strong immunogenicity and the tumor regression could be driven by luciferase-specific T cells. Does the combination therapy (IPI549@HMP plus RT) also effective in wild type CT26 and other tumor types?

Response: Thank you very much for the constructive suggestion. We previously used the luciferase⁺ CT26 cell lines as it allowed us to monitor the recurrence of CT26 tumors postsurgical resection via bioluminescence imaging, as used in many other papers¹⁻⁵. However, we agree the reviewer that the luciferase itself may serve as an antigen. To exclude the effect of potential immunogenicity of luciferase antigens, we repeated the tumor surgical resection study using wild type CT26 cells without luciferase expression. As shown in **Supplementary Fig. 23**, a similarly beneficial effect of the combination therapy (IPI549@HMP plus RT) in suppressing tumor recurrence was observed. We have added this part in the revised manuscript, which is marked in yellow. (Lines 327-330 of Page 15, Revised Manuscript and Page 24, Revised

Supplementary Information).

According to the reviewer's suggestion, we further explored whether such a combined radioimmunotherapy strategy could potentially be used for other tumor types by employing murine B16F10 melanoma as the model in addition to the colon CT26 cancer model. Similar to previous observations, IPI549@HMP+RT also exhibited remarkable therapeutic efficacy against murine postsurgical residual melanoma and significantly reduced the tumor growth rate compared to controls (**Supplementary Fig. 17**). These results successfully proved that our combined radioimmunotherapy strategy in CT26 cancer model can be extended to other types of tumors. (**Lines 287-293 of Page 14, Revised Manuscript and Page 18, Revised Supplementary Information**).

Supplementary Fig. 23 a Schematic illustration of the experiment design to assess the *in vivo* IPI549@HMP-based RT in wild type CT26 tumors. **b** Average tumor growth kinetics of tumor-bearing mice after varied treatments (n = 6). **c** Weight of the excised tumors examined on day 20 after varied treatments (n = 6). **d** Digital photos of the excised tumors examined on day 20 after varied treatments (n = 6). The circle represents cured tumors. SR, surgical resection, RT, radiotherapy. Data were expressed as means \pm SD. Statistical difference was calculated using unpaired student's *t*-test. * $P < 0.05$, ** $P < 0.01$ and *** $P < 0.001$.

Supplementary Fig. 17 IPI549@HMP-augmented RT against postsurgical melanoma. a Schematic illustration of the experiment design to assess the *in vivo* IPI549@HMP-based RT in B16F10 melanoma. **b** Residual tumor growth kinetics of mice after varied treatments (n = 6). **c** Weight of the excised tumors examined on day 15 after varied treatments (n = 6). **d** Digital photos of the excised tumors examined on day 15 after varied treatments. SR, surgical resection, RT, radiotherapy. Data were expressed as means \pm SD. Statistical difference was calculated using unpaired student's *t*-test, ***P<0.001.

Reference:

1. Fang L, *et al.* Engineering autologous tumor cell vaccine to locally mobilize antitumor immunity in tumor surgical bed. *Sci Adv* **6**, aba4024 (2020).
2. Xu J, *et al.* A general strategy towards personalized nanovaccines based on fluoropolymers for post-surgical cancer immunotherapy. *Nat Nanotechnol* **15**, 1043-1052 (2020).
3. Park CG, Hartl CA, Schmid D, Carmona EM, Kim HJ, Goldberg MS. Extended release of perioperative immunotherapy prevents tumor recurrence and eliminates metastases. *Sci Transl Med* **10**, aar1916 (2018).
4. Zhang L, *et al.* In Situ Formed Fibrin Scaffold with Cyclophosphamide to Synergize with Immune Checkpoint Blockade for Inhibition of Cancer Recurrence after Surgery. *Adv Funct*

Mater **30**, 1906922 (2020).

5. Chen Q, *et al.* In situ sprayed bioresponsive immunotherapeutic gel for post-surgical cancer treatment. *Nat Nanotechnol* **14**, 89-97 (2019).

(3) The efficacy of IPI549@HMP treatment depends on DC or MDSC?

Response: Thank you very much for the constructive question. The efficacy of IPI549@HMP treatment is closely related to both DC and MDSC. Localized radiation initiates cell death and the production and release of cytokines as well as chemokines into the tumor microenvironment, which leads to infiltration of tumor-associated DCs (**Supplementary Fig. 19, Lines 307-309 of Page 14-15, Revised Manuscript and Page 20, Revised Supplementary Information**). Then, the immunogenic cell death (ICD) related damage-associated molecular patterns (DAMPs) (**Fig. 5h**) were sensed by tumor-infiltrating DCs, promoting DC activation and effective cross-presentation of tumor-derived antigens to T cells. The stimulated T cells eventually mediate the tumor rejection response.

The tumor microenvironment is populated by various types of inhibitory immune cells including Tregs, alternatively activated macrophages, and myeloid-derived suppression cells (MDSCs), which suppress T cell activation and promote tumor outgrowth¹. MDSCs contribute to tumor progression by promoting the survival of tumor cells, and via stimulation of angiogenesis, tumor-cell invasion of adjacent tissues, and metastasis². In our manuscript, we demonstrate that the unique genomic landscape shaped by surgical resection creates an immunosuppressive milieu characterized by hypoxia and high-influx of myeloid cells, significantly fostering cancer progression and hindering PD-L1 blockade therapy. As depicted in **Fig. 2i, j and m**, MDSCs displayed a substantially increased infiltration in SR tumors. However, we detected that IPI549@HMP-based RT treatment could effectively decrease the frequency of MDSCs compared with those in the other controls (**Fig. 6c, d**). Notably, the ratio of CD8/MDSC, which was well-recognized indicator of antitumor immune balance, was found to be highly improved in IPI549@HMP plus RT group, consistent with the strongest anticancer effect in this group (**Supplementary Fig. 22**).

These data suggested that this IPI549@HMP nanoparticle-based augmented RT process in postoperative CT26 colon tumor mice could not only increase the proportions of positive immune responders (*e.g.* DC) but also suppress those negative immune inhibitors (*e.g.* MDSC), thus eventually establishing an inflamed tumor immunity niche and exerting an effective tumoricidal immune activity that inhibited postsurgical cancer growth. We have clarified this issue in the revised manuscript and highlighted it in yellow. (Lines 316-318 and 322-326 of Page 15, Revised Manuscript and Page 23, Revised Supplementary Information).

Figure S19. Infiltration of dendritic cells in primary tumors. Representative flow cytometric images and relative quantification of dendritic cells (DCs, MHC II⁺CD11c⁺) within tumor tissues 9 days post treatment. G1: control; G2: IPI549@HMP; G3: RT; G4: HMP+RT; G5: IPI549@HMP+RT. Data were expressed as means \pm SD (n = 3). Statistical difference was calculated using unpaired student's *t*-test. **P<0.01.

Supplementary Fig. 22 Quantification by flow cytometry of CD8/MDSC ratios. Data were expressed as means \pm SD (n = 3). Statistical difference was calculated using unpaired student's *t*-test. **P<0.01.

Reference:

1. Hiam-Galvez KJ, Allen BM, Spitzer MH. Systemic immunity in cancer. *Nat Rev Cancer* **21**,

345-359 (2021).

2. Pathria P, Louis TL, Varner JA. Targeting Tumor-Associated Macrophages in Cancer. *Trends Immunol* **40**, 310-327 (2019).

(4) The percentage of TAMs and MDSCs have decreased after combination therapy, Are they killed by CTL?

Response: Thanks very much for your question. The reduction in the percentage of TAMs and MDSCs after combination therapy was mainly caused by inhibition of MDSCs recruitment and repolarization of TAMs into an immunostimulatory phenotype.

Myeloid-derived suppressor cells (MDSCs), characterized by their suppressive effects on immune responses, are important motivators to promote tumor immune escape¹. MDSC-mediated resistance of cancer cells to cytotoxic therapies is a direct process that occurs through promotion of tumor cell survival, and also an indirect process caused by inhibition of T cell responses². Various pairs of chemoattractants and their receptors can stimulate myeloid cell recruitment to tumors³. Moreover, MDSCs possess polarization potentials towards different TAMs with M1 and M2-like phenotypes, depending on the cytokine milieu⁴. Although the classic pro-inflammatory M1-TAM exerted anti-proliferative and cytotoxic activities, the M2-TAM-skewed immunosuppression prevailed in postsurgical TME. Considering the prominent role of immunomodulation on tumor growth, we developed an intelligent HMP-bridged radioimmunotherapy nanoplatform loading with a small molecular PI3-kinase γ (PI3k γ) inhibitor (IPI549) that could overcome immune tolerance by eliminating these immunosuppressive myeloid cells.

In this system, HMP nanoshells possess outstanding catalase activity to decompose endogenous H₂O₂ into O₂, so as to evade hypoxia-mediated MDSCs infiltration and radioresistance. Importantly, IPI549 is capable of reshaping TME by hijacking MDSCs trafficking and switching macrophages from the immunosuppressive M2-like phenotype to the pro-inflammatory M1-like state⁵. Also, local X-ray irradiations not only exert a direct killing effect on tumor cells, but also induce the release of tumor antigens, thus activating innate and

adaptive immune responses and promoting the infiltration of immune cells into TME. These effects interact together to reduce infiltrating myeloid cells within the TME and polarize M2-TAM to M1-TAM, ultimately establishing an inflamed tumor immunity niche that exerts tumoricidal immune activity.

Reference:

1. Talmadge JE, Gabrilovich DI. History of myeloid-derived suppressor cells. *Nat Rev Cancer* **13**, 739-752 (2013).
2. Kumar V, Patel S, Tcyganov E, Gabrilovich DI. The Nature of Myeloid-Derived Suppressor Cells in the Tumor Microenvironment. *Trends Immunol* **37**, 208-220 (2016).
3. Gerhardt T, Ley K. Monocyte trafficking across the vessel wall. *Cardiovasc Res* **107**, 321-330 (2015).
4. Smyth MJ, Ngiow SF, Ribas A, Teng MW. Combination cancer immunotherapies tailored to the tumour microenvironment. *Nat Rev Clin Oncol* **13**, 143-158 (2016).
5. Kaneda MM, *et al.* PI3K γ is a molecular switch that controls immune suppression. *Nature* **539**, 437-442 (2016).

(5) PI3K- γ inhibitor can block neutrophil or MDSC recruitment, does any changes observed in the population in the combination therapy?

Response: Thanks very much for your question. Compared with the control group, we observed a significant reduction of infiltrating MDSCs (CD11b⁺Gr-1⁺CD45⁺) in the treatment group containing PI3K- γ inhibitor, especially in the combination treatment group (**Fig. 6c, d**). Further subdivision of these myeloid cells into neutrophils MDSC (CD11b⁺Ly6G^{high}) and monocytes MDSC (CD11b⁺Ly6C^{high}) revealed that IPI549@HMP mainly impaired neutrophils MDSC recruitment, while the combination treatment reduced both neutrophils MDSC and monocytes MDSC infiltration by 57% and 53%, respectively. These results suggested that our IPI549@HMP plus RT strategy can efficiently hinder MDSC recruitment. We have clarified this

issue in the revised manuscript and highlighted it in yellow. (Lines 311-318 of Page 15, Revised Manuscript and Page 22, Revised Supplementary Information)

Supplementary Fig. 21 Infiltration of MDSC subpopulations in primary tumors.

Representative flow cytometric images and relative quantification of monocytes MDSC (CD11b⁺Ly6C^{high}) and neutrophils MDSC (CD11b⁺Ly6G^{high}) within tumor tissues 9 days post treatment. G1: control; G2: IPI549@HMP; G3: RT; G4: HMP+RT; G5: IPI549@HMP+RT. Data were expressed as means \pm SD (n = 3). Statistical difference was calculated using unpaired student's *t*-test. *P<0.05 and **P<0.01.

(6) Which subsets of T cells were necessary for tumor control? CD8⁺T cells or CD4⁺T cells? Do NK cells also essential for tumor control in combination therapy?

Response: Thank you very much for this insightful question, which is highly appreciated. Based on the reviewer's valuable suggestion, we performed CD8⁺ T cell and CD4⁺ T cell depletion experiments on a postsurgical colon cancer mouse model to examine which T cell subtype plays a key role in tumor control. As shown in **Supplementary Fig. 23**, we observed that IPI549@HMP + RT treatment lost most of the immunotherapeutic effect in primary CT26 tumors after CD8⁺ T cells depletion. However, the combined radioimmunotherapy still greatly inhibited tumor growth after CD4⁺ T cells depletion. These results indicated that CD8⁺ T cells

deeply involved in IPI549@HMP mediated radiation sensitization and immunotherapeutics. Meanwhile, we found a significant increment of natural killer (NK) cells infiltration within the tumor site after combination therapy by NKp46 immunofluorescence staining (**Supplementary Fig. 24**), indicating that IPI549@HMP+RT could augment innate and adaptive immune responses against tumors, thereby decreasing immunosuppression and potentiating the responsiveness of tumors to radiation. We have clarified this issue in the revised manuscript and highlighted it in yellow. (**Lines 330-340 of Page 15-16, Revised Manuscript and Page 24-25, Revised Supplementary Information**).

Supplementary Fig. 23 **a** Schematic illustration of the experiment design to assess the *in vivo* IPI549@HMP-based RT in wild type CT26 tumors. **b** Average tumor growth kinetics of tumor-bearing mice after varied treatments (n = 6). **c** Weight of the excised tumors examined on day 20 after varied treatments (n = 6). **d** Digital photos of the excised tumors examined on day 20 after varied treatments (n = 6). The circle represents cured tumors. SR, surgical resection, RT, radiotherapy. Data were expressed as means \pm SD. Statistical difference was calculated using unpaired student's *t*-test. *P<0.05, **P<0.01 and ***P<0.001.

Supplementary Fig. 24 NKp46 expression within primary tumor after varied treatment.

Immunofluorescence images and relative quantification of tumor slices stained with NKp46 antibody after varied treatments as indicated. Data were denoted as the mean \pm SD (n = 3). Statistical difference was calculated using unpaired student's *t*-test. *P<0.05.

Response to Reviewer 3

Comments from Reviewer #3 (expertise: Radioimmunotherapy):

In this manuscript, Guan et al. developed a PI3K γ -targeted nanosystem, established an experimental colon cancer model for post-surgical immuno-radiotherapy, and demonstrated a robust tumor control and immune memory. The work is original and of significance to its field. The experiments are adequate and well-designed. The methodology is well-structured and supported with comprehensive visuals, leading up to comprehensible and proper conclusions, although some of the analysis could use more expounding and details.

Response: Thank you very much for the kind comment and suggestion. Please find the following detailed responses. We greatly appreciate the positive comments and address major concerns below.

(1) However, I would like to elaborate on one major concern I am left with and offer two suggestions. My major concern lies in the establishment of the surgical resection (SR) model system, which is the foundation of this study. In Figure 1, the authors compared post-SR tumors mainly with untreated tumors. Fig. 1b showed significant larger volumes of post-SR tumors on day 20 to day 24, as compared to untreated or sham operation tumors. It is unclear, however, how the authors adjust the start point of the growth curve among three treatment groups. On day 10, for example, the post-SR residual tumors are of a smaller size (50-80 mm³) than the untreated tumors (150-200 mm³), which is not reflected in Fig. 1b. On page 7, the authors stated “..compared with size-matched tumors (line 119)”. To match the size of the tumor in different treatment groups, the curves must start at different time (dates) due to the surgical removal of a large proportion of the tumor in the SR group, which is, again, not reflected in Fig. 1b. It is critically important to clarify (in “Methods”, “Results”, and “Figure Legends”) the start point of comparison, as size-matching may result in an inappropriate comparison between a late-stage faster-growing post-SR tumors with early-stage slower-growing untreated tumors. As the follow-up experiments in Fig. 1d-1q implemented the same comparison strategy, an inappropriate design or a lack of justification regarding the comparison strategy leads to

questioning the credibility of all these transcriptomics and immune profiling data.

Response: Thank you very much for the constructive suggestion, which is highly appreciated. We are very sorry for the inappropriate description and sincerely thankful for your advice. We have inserted the detailed treatment information in the revised manuscript (including the treatment starting time and the start point of comparison) (**Lines 499-503 of Page 23-24, Revised Manuscript**). Indeed in this study, the starting point for comparison between the control and SR groups was not size-matched but time-matched. We regret that we previously did not record the tumor size of each group before the surgical treatment was performed, making the tumor volume change caused by resection not clearly reflected in the figure of tumor-growth kinetics. To improve the credibility of the manuscript, we repeated this part of experiment again in this revision round and listed the experimental results in detail below, which we hope to meet your approval.

To determine the effects of SR on modulating tumorigenesis, a postsurgical CT26 colon mouse model system, in which SR was performed to partially remove the tumor surgically, was introduced in this study. Concurrently, sham operation was performed on the contralateral flank of tumor-bearing mice to determine whether the normal tissue injury caused by surgery can also lead to rapid tumor progression (**Fig. 2a**). We found that compared with tumors in untreated and sham operation groups, SR-treated mice initially decreased the size of the treated tumors. However, the residual tumors eventually grew larger than the tumors in other groups. (**Fig. 2b, c and Supplementary Fig. 1**). Meanwhile, no obvious difference was observed between the sham operation and untreated groups. These results together suggest that the presence of residual tumor following SR induced the accelerated cancer progression. (**Lines 100-103 and 106-108 of Page 5, Revised Manuscript and Page 2, Revised Supplementary Information**).

Fig.2 SR-driven immunosuppression accelerates local tumor progression. **a** Schematic illustration of surgical resection (SR) treatment. 1×10^6 CT26 cells were subcutaneously injected into the right flank of BALB/c mice. SR or sham operation was conducted on the right tumor or left skin, respectively, on day 10 post-inoculation. **b** Residual tumor growth kinetics of mice in Untreated, SR and Sham operation groups. ($n = 6$). **c** Weight of the excised tumor on day 20 after varied treatments ($n = 6$). SR, surgical resection. Data were expressed as means \pm SD. Statistical difference was calculated using unpaired student's *t*-test. ns, not significant, * $P < 0.05$ and *** $P < 0.001$.

Supplementary Fig. 1 SR accelerates local tumor progression. **a** Representative digital photos of the treated mice. The arrow indicates the surgical site, and the circle represents the tumor site. **b** Digital photos of the excised tumors on day 20 after varied treatments. SR, surgical resection.

(2) My first suggestion is in relation to my stated concern. To establish post-SR tumors as the specific target to benefit from the IPI549@HMP+RT regimen, the authors would better compare post-SR tumors with untreated tumors for their responses to IPI549@HMP+RT. In other words, I suggest that experiments in Figure 5 be repeated with non-SR (untreated) tumors to strengthen the conclusion that post-SR tumors exhibit a better response to IPI549@HMP+RT than non-SR (untreated) tumors. In these new experiments, the author would have to clarify the treatment starting time for both groups, i.e., whether to match the tumor size or the time after tumor inoculation (tumor-growth phase).

Response: Thank you very much for the constructive suggestion, which is highly helpful in improving the quality of the manuscript. Based on the reviewer's suggestion, we added a new trial of no-SR tumors treated with IPI549@HMP+RT in a time-matched condition after tumor inoculation for comparison. Although the combination therapy indeed inhibits tumor growth to some extent in the no-SR group, the tumor response rate remains lower than that of the SR-treated group (**Supplementary Fig. 23**). This demonstrates the higher specificity and better efficacy of our postsurgical targeted therapy. We have clarified this issue in the revised manuscript and highlighted it in yellow. (**Page 24, Revised Supplementary Information**).

Supplementary Fig. 23 **a** Schematic illustration of the experiment design to assess the *in vivo* IPI549@HMP-based RT in wild type CT26 tumors. **b** Average tumor growth kinetics of tumor-bearing mice after varied treatments (n = 6). **c** Weight of the excised tumors examined on day 20 after varied treatments (n = 6). **d** Digital photos of the excised tumors examined on day 20 after varied treatments (n = 6). The circle represents cured tumors. SR, surgical resection, RT, radiotherapy. Data were expressed as means \pm SD. Statistical difference was calculated using unpaired student's *t*-test. *P<0.05, **P<0.01 and ***P<0.001.

(3) My second suggestion is related to the abscopal effect of IPI549@HMP+RT (Fig. 7). Fig. 7c

showed no inhibitory effects of anti-PD-L1 alone, whereas IPI549@HMP+RT+aPDL1 elicited a robust inhibition on either primary or abscopal tumors. These results suggest that IPI549@HMP+RT creates an enhanced vulnerability to anti-PD-L1, likely due to an induction of PD-L1 expression. I recommend a comparison of PD-L1 expression in both primary and abscopal tumors before and after IPI549@HMP+RT treatment.

Response: Thanks very much for your valuable counsel, which is highly helpful in improving our manuscript. Generally, PD-L1 expression on tumor cells was a potent predictor of responses to anti-PD-L1 therapy. To investigate whether IPI549@HMP+RT treatment induces PD-L1 expression upregulation in both primary and distant tumors, we isolated tumor tissues for PD-L1 immunofluorescence staining before and 5 days after treatment, respectively. As depicted in **Supplementary Fig. 29**, PD-L1 expression level was found to be increased in both primary and distant tumors after IPI549@HMP+RT treatment. This upregulated PD-L1 expression may provide an opportunity for PD-L1 blockade that would uncover the full cytotoxic potential of host immunity against tumors. We have clarified this issue in the revised manuscript and highlighted it in yellow. (**Lines 372-375 of Page 17-18, Revised Manuscript and Page 30, Revised Supplementary Information**).

Supplementary Fig. 29 PD-L1 expression in primary and distant tumors. Immunofluorescence and relative quantification of PD-L1 expression in mice before and 5 days after IPI549@HMP+RT treatment. RT, radiotherapy. Data were expressed as means \pm SD (n = 3). Statistical difference was calculated using unpaired student's t-test. *P<0.05.

Reviewers' Comments:

Reviewer #1:

Remarks to the Author:

The author has well answered all questions and response raised by reviewers. The paper is generally suitable for publication.

Reviewer #2:

None

Reviewer #3:

Remarks to the Author:

My concerns have been fully addressed. The authors performed additional experiments according to my suggestions, and the results strengthened their conclusions.

Response to Reviewers 1

Comments from Reviewer #1:

The author has well answered all questions and response raised by reviewers. The paper is generally suitable for publication.

Response: Thank you very much for the positive comments and constructive suggestions.

Response to Reviewer 3

Comments from Reviewer #3 :

My concerns have been fully addressed. The authors performed additional experiments according to my suggestions, and the results strengthened their conclusions.

Response: Thank you very much for the positive comments and constructive suggestions.